# Anatomically and functionally distinct thalamocortical inputs to primary and secondary mouse whisker somatosensory cortices

Sami El-Boustani [1,3,4 ✉], B. Semihcan Sermet[1,4], Georgios Foustoukos[1], Tess B. Oram[2], Ofer Yizhar [2] & Carl C. H. Petersen [1 ✉]

Subdivisions of mouse whisker somatosensory thalamus project to cortex in a region-specific and layer-specific manner. However, a clear anatomical dissection of these pathways and their functional properties during whisker sensation is lacking. Here, we use anterograde trans-synaptic viral vectors to identify three specific thalamic subpopulations based on their connectivity with brainstem. The principal trigeminal nucleus innervates ventral posterior medial thalamus, which conveys whisker-selective tactile information to layer 4 primary somatosensory cortex that is highly sensitive to self-initiated movements. The spinal trigeminal nucleus innervates a rostral part of the posterior medial (POm) thalamus, signaling whisker-selective sensory information, as well as decision-related information during a goal-directed behavior, to layer 4 secondary somatosensory cortex. A caudal part of the POm, which apparently does not receive brainstem input, innervates layer 1 and 5A, responding with little whisker selectivity, but showing decision-related modulation. Our results suggest the existence of complementary segregated information streams to somatosensory cortices.

[1] Laboratory of Sensory Processing, Brain Mind Institute, Faculty of Life Sciences, Ecole Polytechnique Fédérale de Lausanne (EPFL), EPFL-SV-BMI-LSENS Station 19, CH-1015 Lausanne, Switzerland. [2] Department of Neurobiology, Weizmann Institute of Science, 234 Herzl Street POB 26, 7610001 Rehovot, Israel. [3] Present address: Department of Basic Neurosciences, Faculty of Medicine, University of Geneva, 1 Rue Michel-Servet, 1206 Geneva, Switzerland. [4] These authors contributed equally: Sami El-Boustani, B. Semihcan Sermet. ✉email: sami.el-boustani@unige.ch; carl.petersen@epfl.ch

The thalamus is a group of nuclei located in the center of the brain, which provide important excitatory glutamatergic input to all regions of the cortex. Sensory information is relayed through parallel modality-specific thalamic nuclei to modality-specific sensory cortices for vision, hearing, taste and touch. For rodents, whisker-related tactile somatosensation provides important information about the structure of their immediate surroundings, and several whisker-related thalamocortical signaling pathways have begun to be characterized[1–11]. In particular, the lemniscal and paralemniscal pathways involve the ventral posterior medial (VPM) nucleus and the posterior medial (POm) group of the thalamus, respectively, with segregated thalamocortical projections[11–14]. It has long been hypothesized that these pathways convey distinct tactile information to the cortex in awake behaving animals. However, description of these thalamic networks has been challenging due to the sensitivity of their activity to brain state and their apparent heterogeneity. Indeed, several lines of research have accumulated evidence indicating that POm might be composed of two populations separated along the rostro-caudal axis with distinct thalamocortical axonal projections[15,16] and distinct synaptic inputs—either from both cortical and brainstem origin or from cortical origin only[17–19]. Moreover, genetic evidence suggests a corresponding molecular heterogeneity of this nucleus[20]. By using a novel adeno-associated viral (AAV) vector-based circuit mapping approach[21], we undertook to dissect thalamic circuits responsible for conveying tactile information to the cortex and probe the activity of these thalamocortical pathways in awake behaving mice. We report three distinct thalamocortical pathways to mouse whisker primary and secondary somatosensory cortices, each carrying different sensorimotor signals.

## Results

**AAV-mediated dissection of whisker-related thalamic nuclei.** We first sought to isolate the thalamic region receiving direct inputs from the trigeminal nucleus principalis (Pr5) in the brainstem. Using an AAV vector with anterograde trans-synaptic transfection properties[21], we expressed Cre-recombinase in the thalamus through AAV injection in Pr5. A second AAV injection of a Cre-dependent fluorescent protein (tdTomato) construct in the thalamus revealed a population of neurons within the VPM nucleus projecting specifically to the whisker primary somatosensory cortex (wS1) (Fig. 1a), as expected for the well-characterized lemniscal sensory pathway[1]. We then performed the same experiment, this time targeting the interpolaris division of spinal trigeminal nucleus (Sp5i) known to innervate the POm group of the thalamus as part of the paralemniscal pathway[1]. This revealed neurons in the most anterior part of POm that project mainly in the whisker secondary somatosensory cortex (wS2) (Fig. 1b). Similar results were found when small injections were targeted to rostral Sp5i ($n = 5$ mice). To identify the complementary POm neuronal population, which did not express Cre-recombinase through anterograde trans-synaptic transfection from the brainstem, we delivered an AAV vector to express the enhanced yellow fluorescent protein (eYFP) in a Cre-OFF manner in the thalamus (see "Methods"). We observed expression of eYFP in the posterior part of POm with broad axonal innervation of the cortex (Fig. 1c). Further analysis showed that the two main trigemino-thalamo-cortical circuits going through VPM and POm —defined here as first-order (FO) nuclei—mainly target layer 4 (L4) of the cortex whereas the higher-order (HO) subdivision of POm targets layer 5A (L5A) and layer 1 (L1) (Fig. 1d, e). Note that expression of eYFP in POm-HO could potentially include neurons from POm-FO if not enough neurons from Sp5i were transfected from injections in the brainstem. However, alignment of brain

slices and two-photon tomography data to a reference atlas[22] helped identify an anatomical boundary between POm-FO and POm-HO along the rostro-caudal axis with little overlap between these neuronal populations (Supplementary Fig. 1, Supplementary Video 1; POm-FO approximately centered at: −1.7 mm posterior, 1.2 mm lateral, 3 mm deep relative to bregma; POm-HO approximately centered at: −2.2 mm posterior, 1.2 lateral, 2.7 mm deep relative to bregma). In addition, injection of cholera-toxin subunit B (CTB) conjugated with Alexa647 in POm-HO to retrogradely label neurons projecting to this nucleus suggests that POm-HO receives inputs from cortical neurons in both L5 and L6[23] (Supplementary Fig. 2). Thus POm-FO and POm-HO subdivisions correspond anatomically to the convergence (Sp5i and cortical inputs) and non-convergence (cortical inputs only) zones previously reported[17,18]. In summary, here we define three distinct thalamocortical pathways that innervate the cortex in a region-specific and layer-specific manner (Fig. 1f).

**Synaptic integration of thalamocortical projections in wS2.** Although synaptic integration of thalamocortical inputs in wS1 has begun to be characterized[8,9,11,24,25], little is known regarding synaptic responses of wS2 excitatory neurons to thalamocortical axonal stimulation. To characterize glutamatergic drive exerted by POm-FO and POm-HO thalamocortical circuits in different layers of wS2, we performed ex vivo whole-cell recordings of membrane potential in parasagittal slices of mice expressing channelrhodopsin-2 in these nuclei (Fig. 2a). We followed the same AAV-based strategy as before, expressing channelrhodopsin-2 in these two nuclei by using Cre-ON or Cre-OFF vectors in the thalamus after injection of the trans-synaptic anterograde AAV1.CaMKIIa.Cre viral vector in Sp5i. We targeted excitatory neurons across all layers (Fig. 2b) and recorded monosynaptic excitatory postsynaptic potentials (EPSPs)[24] in response to 1 ms blue light pulses applied in wS2 to pathway-specific thalamic axons expressing ChR2 (Fig. 2c) (see "Methods"). By recording from many neurons in the same brain slice, we measured the amplitude of the EPSPs in neurons located in different cortical layers evoked in response to the same stimulation of pathway-specific thalamic axons. In order to better compare across different slices, which could contain varying expression levels of ChR2, we normalized the layer-specific responses to the main input layer as defined by thalamocortical axonal innervation. This strategy allowed us to map the profile of monosynaptic thalamic input in the cortex across layers. For POm-FO axons, we observed weaker responses in superficial layers compared with deeper layers with dominant synaptic inputs in the main input layer L4 (Fig. 2d, ANOVA test $p = 0.02$, Kruskal–Wallis test comparing L4 with all other layers, $p = 0.0012$). POm-HO thalamocortical inputs elicited broad responses across layers with a dominant input in L5A pyramidal cells located where the main axonal innervation was observed (Fig. 2e–g, ANOVA test $p = 0.018$, Kruskal-Wallis test comparing L5A with all other layers, $p = 0.0007$).

**Distinct thalamocortical properties during passive stimuli.** Taking advantage of the versatility of this trans-synaptic viral approach, we next characterized the functional properties of thalamocortical projections originating from VPM-FO, POm-FO, and POm-HO subnuclei. In order to characterize the activity of the three thalamocortical pathways in vivo, we expressed the genetically encoded calcium indicator GCaMP6s using the dual injection strategy described in Fig. 1. Axons were imaged using a two-photon microscope through a microprism window assembly[26,27] to access deep cortical layers in wS1 or wS2 (Fig. 3a–c, Supplementary Video 2). To image POm-HO axons,

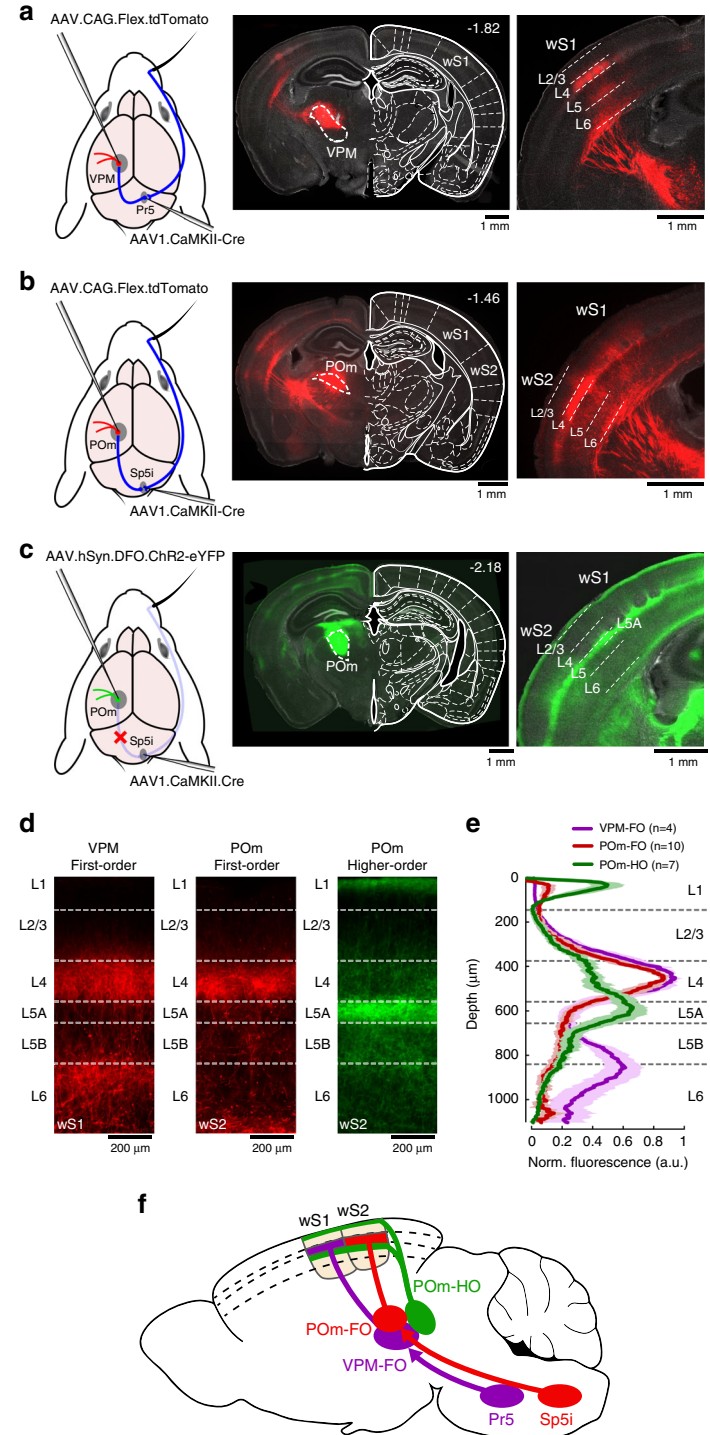

**Fig. 1 Whisker somatosensory thalamic nuclei and their cortical projections revealed through AAV-mediated anterograde trans-synaptic gene expression. a** AAV1 viral vector was injected in Pr5 of the brainstem to express Cre-recombinase in a trans-synaptic anterograde manner. A second AAV injection in the thalamus expressing a Cre-dependent tdTomato fluorescent protein resulted in labeling of VPM neurons receiving direct inputs from Pr5. Left: schematic of the injection protocol. Middle: Example coronal section with VPM neurons expressing tdTomato in comparison to a reference atlas[22] (distance from bregma indicated). Right: Axonal innervation of VPM neurons in wS1. This experiment was repeated in four mice with similar results. **b** Same as **a**, but for POm neurons receiving direct inputs from Sp5i. This experiment was repeated in ten mice with similar results. **c** Same as **b**, but for POm neurons not expressing Cre-recombinase through trans-synaptic transfection from Sp5i injections. Here, the second viral vector injected in the thalamus only allowed expression of eYFP conditionally on the absence of Cre-recombinase. This experiment was repeated in seven mice with similar results. **d** Examples of laminar-specific axonal innervation in somatosensory cortices originating from different thalamic nuclei. **e** Normalized fluorescent expression profile averaged over mice ($n = 4$ mice for VPM first-order (VPM-FO), $n = 10$ mice for POm first-order (POm-FO), $n = 7$ mice for POm higher-order (POm-HO)). Shaded areas: s.e.m. **f** Schematic of the different somatosensory thalamocortical circuits. The schematic drawings of the brain in panels **a**–**c** are reproduced from Paxinos and Franklin (2001) with permission from Elsevier.

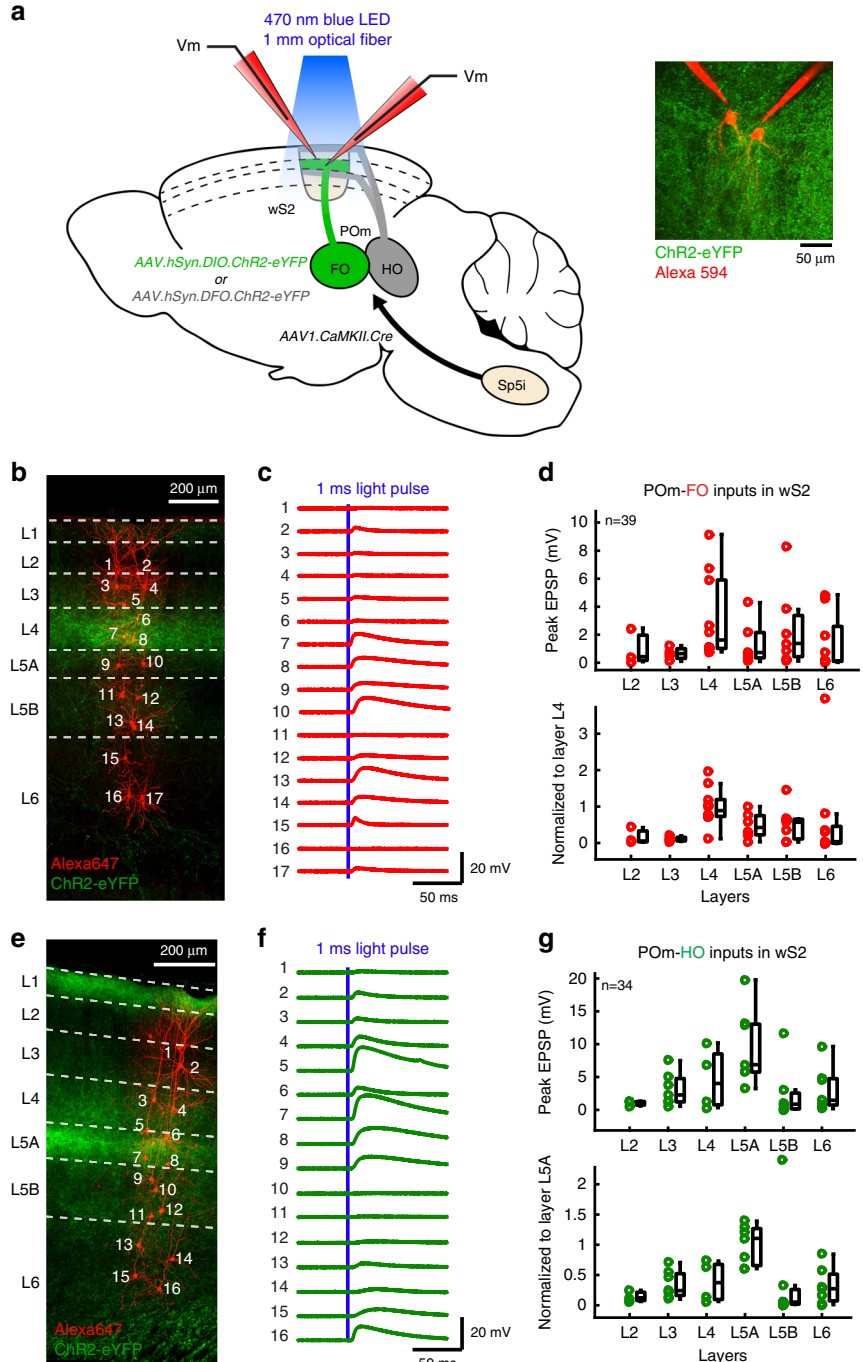

**Fig. 2 Ex vivo whole-cell recordings in brain slices of POm-FO and POm-HO inputs to excitatory neurons across layers in wS2. a** Schematic showing the strategy used to selectively express ChR2-eYFP in FO or HO subdivisions of POm. Method used to activate thalamocortical axons expressing channelrhodopsin-2 in order to evoke postsynaptic potentials in the somatosensory cortex is illustrated. A 470 nm wavelength light was delivered with a LED light source coupled with a 1 mm optic fiber onto wS2. Inset: Two-photon microscopy image of an in vitro whole-cell patch–clamp recording of two neurons filled with Alexa 594. **b** Confocal z-projection of wS2 in a parasagittal slice after fixation. ChR2-eYFP was expressed in POm-FO axons, and recorded neurons were filled with biocytin followed by staining with streptavidin conjugated to Alexa 647. This experiment was repeated in three mice with similar results. **c** Light-evoked excitatory postsynaptic potentials (EPSPs) from recorded neurons labeled in **b** following 1 ms light pulses. **d** Top: Peak amplitude of EPSPs evoked by optogenetic stimulation of POm-FO axons recorded in cortical excitatory neurons ($N = 3$ mice, $n = 39$ neurons) across different layers in wS2. On each box, central mark indicates the median and edges indicate 25th and 75th percentiles. The whiskers extend from the minimum data point comprised within 1.5× of the interquartile range to the 25th percentile and from the maximum data point comprised within 1.5× of the interquartile range to the 75th percentile. Bottom: same responses normalized to the average peak EPSP recorded in the main input layer (L4) for each experiment. **e–g** Same as **b–d** but for POm-HO axon stimulation during whole-cell recording of neurons in wS2 ($N = 3$ mice with similar results, $n = 34$ neurons). Here, EPSPs were normalized to the average peak EPSPs from L5A neurons for each experiment.

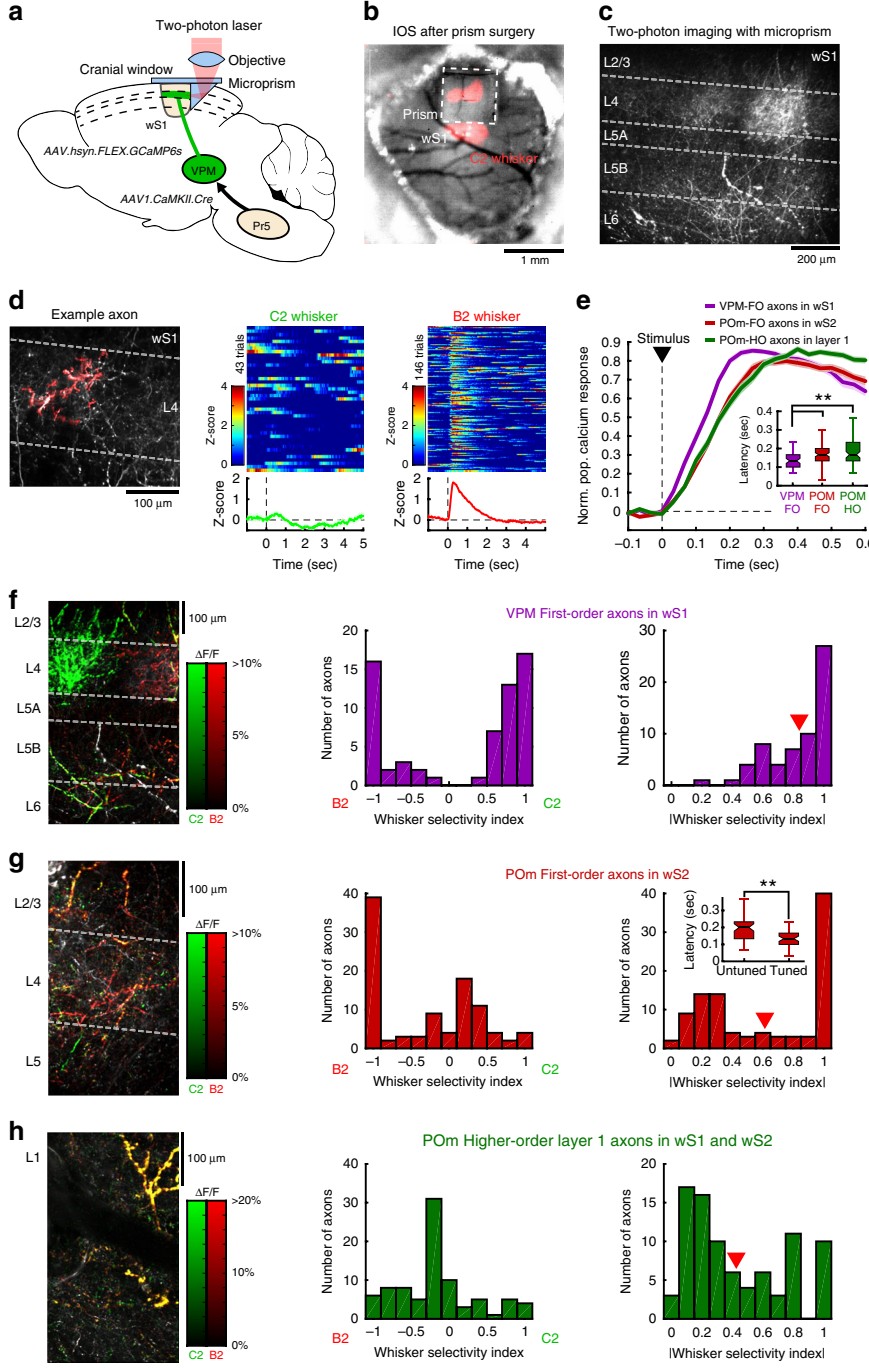

**Fig. 3 Calcium imaging of thalamocortical axons during passive whisker stimulation. a** Schematic of viral vector injection and microprism implantation for in vivo calcium imaging of thalamocortical axons expressing GCaMP6s. **b** Intrinsic optical signal in wS1 during whisker stimulation. This experiment was repeated in 14 mice with similar results. **c** Two-photon image of VPM axons in wS1. Similar results were observed in three mice. **d** Left: Example of a region of interest for axonal segments with high calcium signal correlation in wS1 (red). Right: Corresponding calcium responses to C2 or B2 whisker stimulation in z-score across trials. The average calcium responses are shown below. **e** Normalized calcium responses averaged over all axons for each population (3 mice, n = 62 axons for VPM-FO; 3 mice, n = 101 axons for POm-FO; 8 mice, n = 86 axons for POm-HO). Inset: Response latency comparison (Kruskal–Wallis two-sided test with Bonferroni correction, **p = 0.008 for VPM-FO vs. POm-FO, **p = 0.004 for VPM-FO vs. POm-HO, p = 1 for POm-FO vs. POm-HO). Boxplot: central mark indicates the median and edges indicate 25th and 75th percentiles. Whiskers extend to the largest or smallest point comprised within 1.5× of the interquartile range from both edges. **f** Left: Two-photon image of the calcium response ($\Delta F/F$) for C2 (green) and B2 (red) whisker stimulation overlaid on top of VPM axonal innervation (gray) in wS1. Middle: Distribution of whisker selectivity indices for the corresponding axonal population (3 mice, n = 62 axons). Right: Distribution for the absolute value of the whisker selectivity index. Red arrow: average value. **g** Same as **f**, but for POm-FO axons in wS2 (3 mice, n = 101 axons). Inset in the right panel: Response latency comparison between tuned (|WSI| ≥ 0.75) and untuned (|WSI| < 0.75) axons (Kruskal–Wallis two-sided test, **p = 0.002). Boxplot statistics same as in (**e**) inset. **h** Same as **f**, but for putative POm-HO axons located in layer 1 (8 mice, n = 86 axons). Kruskal–Wallis test with Bonferroni correction comparing |WSI|: p = 0.003 for VPM-FO vs. POm-FO; p = 4 × 10⁻¹¹ for VPM-FO vs. POm-HO; and p = 4 × 10⁻¹¹ for POm-FO vs. POm-HO.

we took advantage of the laminar segregation of POm-FO and POm-HO inputs and imaged axons in layer 1 of Gpr26-Cre mice expressing GCaMP6s specifically in POm (see "Methods"). We first focused on axonal responses following passive stimulation of the C2 whisker or its adjacent B2 whisker. Highly correlated axonal segments, presumably originating from the same neuron, were isolated in the image and the corresponding calcium signal was extracted and z-scored (Fig. 3d, see "Methods"). Calcium responses to whisker stimuli were typically fast, transient and reliable across trials (Fig. 3d) with VPM-FO responses displaying a shorter latency compared to POm-FO and POm-HO axons (Fig. 3e).

Comparing the spatial distribution of axons relative to their whisker selectivity, we observed that VPM-FO neurons form highly segregated axonal domains with strong whisker-specific preferences in the corresponding L4 barrel (Fig. 3f). In contrast, POm-FO axons were more intermingled with no clear anatomical domains and with mixed weak and strong whisker selectivity (Fig. 3g). We quantified the differences in the spatial distribution of highly tuned axons for C2 and B2 whiskers using an index of overlap (see "Methods"). Comparing between VPM-FO and POm-FO thalamocortical projections, we found significantly more overlap between C2- and B2-responding axons in POm-FO than VPM-FO (VPM-FO: $n = 19$ fields of view, $0.023 \pm 0.014$ mean $\pm$ s.e.m.; POm-FO: $n = 16$ fields of view, $0.217 \pm 0.071$ mean $\pm$ s.e.m., Kruskal–Wallis unpaired test, $p = 0.0063$). POm-HO axons imaged in L1 mainly displayed sensory responses that were relatively unselective (Fig. 3h) consistent with previous papers reporting broader receptive fields for non-lemniscal pathways such as POm compared to VPM neurons in urethane-anesthetized rats[3,28]. POm-FO axons displayed whisker-specific responses and were significantly more tuned than POm-HO. Interestingly, POm-FO axons with strong whisker preference displayed significantly shorter response latencies than axons with weak whisker preference (Fig. 3g, right), potentially highlighting two neuronal populations driven dominantly by Sp5i (tuned, short latency resembling VPM-FO) or by cortex (untuned, longer latency resembling POm-HO) as previously suggested[19].

**Pathway-specific responses during sensorimotor behaviors.** Next, we studied the responses of these axonal populations during whisker-based goal-directed behaviors. It has been hypothesized that motor activity and brain state can differentially affect lemniscal and paralemniscal pathways[29–33]. To address this question, we trained water-restricted mice in a two-whisker discrimination task (Fig. 4a). To tease apart signals related to motor outputs from potential reward signals, water was delivered only if mice licked a spout upon C2 whisker stimulation but not for B2 whisker. As a result, mice learned to lick preferentially in response to C2 whisker deflection (Fig. 4a). The first lick reaction time was $0.29 \pm 0.18$ s (mean $\pm$ SD). Calcium responses of thalamic axons to whisker stimulation were enhanced in trials where the mouse licked the spout compared to no-lick trials (Fig. 4b, c, Supplementary Fig. 3). Although VPM-FO axons exhibited much stronger and more prolonged responses in lick trials compared to no-lick trials, this difference was not significant during an early pre-reaction time phase of 0.266 s, roughly corresponding to the time-to-peak for the no-lick condition (Fig. 4d). This indicates that the decision to lick or not to lick the spout did not influence the early response of VPM-FO neurons, consistent with reliable tactile coding in the lemniscal primary sensory thalamus, relatively invariant to subjective report, in agreement with a previous study[34]. In contrast, POm-FO and POm-HO displayed an enhanced transient response during lick trials that was

significantly stronger than in the no-lick condition, even during the early pre-lick period (Fig. 4e, f, Supplementary Fig. 3). This analysis was robust for different time windows (see "Methods"). Our results suggest a potential role for POm-FO and POm-HO in decision-making, possibly resulting from corticothalamic inputs (Supplementary Fig. 2).

We then focused on the prolonged calcium responses found for VPM-FO axons that seem to correlate with licking, which is also often associated with other facial and body movements. Indeed, individual VPM axons displayed a distribution of calcium response latencies that varied with first lick reaction time (Fig. 4b). This was even more apparent in the response for the non-preferred whisker where strong calcium transients were evoked during lick events exclusively. Whisker movements typically correlate with licking[35], and licking responses in VPM axons are likely at least in part due to whisking-related increases in VPM activity[31,36]. In contrast, individual POm-FO axons responded to passive whisker stimuli with enhanced amplitude during lick trials as compared to no-lick trials, appearing as a form of gain modulation (Fig. 4c). As a result, these axons conserved their whisker selectivity regardless of the motor output. In addition, calcium transients in POm-FO and POm-HO axons display no strong timing correlation with first lick reaction time, in contrast to VPM-FO axons (Fig. 4g, Kruskal–Wallis test with Bonferroni correction, $p = 2 \times 10^{-6}$ for VPM-FO vs. POm-FO, $p = 4 \times 10^{-6}$ for VPM-FO vs. POm-HO, $p = 1$ for POm-FO vs. POm-HO). Thus, VPM-FO axons are more excited during licking compared to POm-FO and POm-HO, which was also apparent when looking at calcium signals evoked by isolated spontaneous lick events or calcium activity decay following the offset of licking bouts in VPM-FO axons as compared to POm-FO and POm-HO axons (Supplementary Fig. 4). As a result, calcium responses to self-initiated facial movements prevented VPM-FO axons from conserving information about the passive whisker stimulus as reflected in the dramatic drop in absolute whisker selectivity index (Fig. 4h). In contrast, POm-FO axons did not display a significant decrease in selectivity and became the thalamocortical inputs with the highest whisker selectivity during lick trials (Fig. 4h). Subdividing POm-FO axons into highly tuned and untuned populations revealed some resemblance of the tuned axons with VPM-FO axons although influence of licking on response properties was much weaker and both tuned and untuned axons displayed a strong modulation in the early response phase (Supplementary Fig. 5), indicating an important contribution of top-down inputs not seen in VPM-FO axons.

## Discussion

Using a viral vector mediated gene delivery approach, we were able to express anatomical markers, opsins and calcium indicators in specific thalamic populations, allowing us to investigate the function of these thalamocortical circuits at the level of synaptic integration in cortical neurons ex vivo, as well as in awake quiet or behaving mice. We found that tactile information conveyed by three distinct thalamic projections to somatosensory cortices differ in terms of their whisker selectivity, sensitivity to self-initiated movements and modulation during decision-making in a goal-directed task. Our results suggest that these complementary encoding properties might act in concert at the cortical level to mediate behavior-dependent and -independent representation of tactile scenes.

Using the recent finding that AAV serotype 1 displays strong anterograde trans-synaptic transfection properties[21], we were able to identify whisker-related thalamic nuclei based on their input from the brainstem. In particular, this AAV-mediated dissection of thalamic nuclei revealed two parallel trigemino-thalamo-

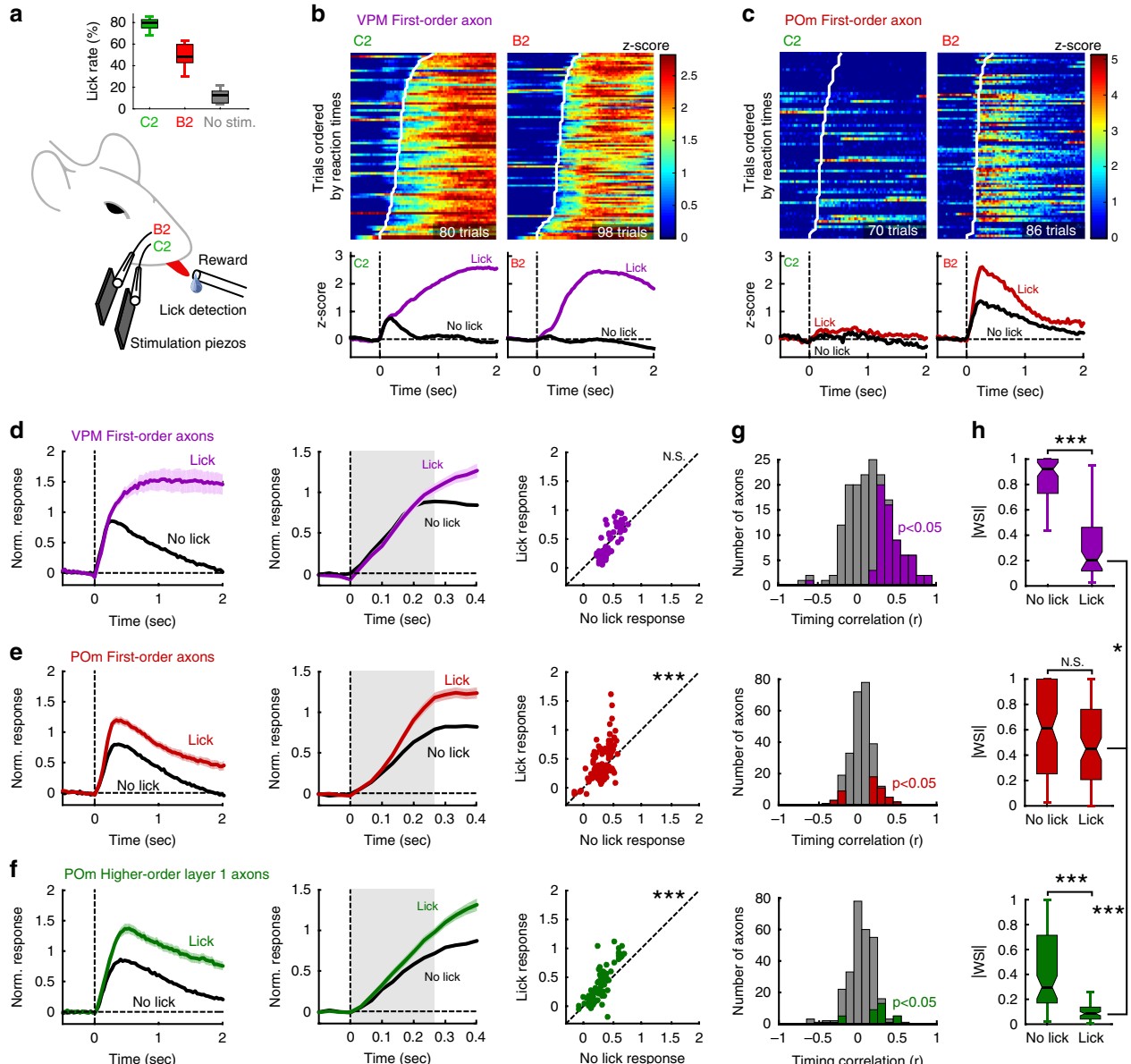

**Fig. 4 Distinct sensory information in parallel thalamocortical pathways during goal-directed behavior. a** Schematic of a mouse performing a two-whisker discrimination task, and the average lick rate over all imaging sessions (>4 days of training) for all stimulus conditions across mice (mean lick rates over $n = 14$ mice, Kruskal–Wallis test with Bonferroni correction: $p = 0.02$ for C2 vs. B2; $p = 2 \times 10^{-8}$ for C2 vs. No stim; and $p = 0.009$ for B2 vs. No stim). Boxplot: central mark indicates the median and edges indicate 25th and 75th percentiles. Whiskers extend to the largest or smallest point comprised within 1.5× of the interquartile range from both edges. **b** Calcium responses (z-score) for an example VPM-FO axon during lick trials upon C2 or B2 whisker stimulation. Trials are ordered according to lick reaction times, which are shown with a white line on color maps. Average responses are shown below for lick and no-lick conditions. **c** Same as **b**, but for a POm-FO axon. **d** Left: Calcium responses averaged over all VPM-FO axons with significant responses to whisker stimuli during lick and no-lick trials, normalized to the no-lick condition. Dark lines: mean value and shaded areas: s.e.m. Middle: Early phase of the response over the first 0.4 s. Right: Comparison of the response amplitude between lick and no-lick conditions averaged over gray area (0–0.266 s) in middle panel (Wilcoxon paired two-sided test, $p = 0.94$, N.S. not significant). **e, f** Same as **d**, but for POm-FO axons (***$p = 3 \times 10^{-5}$ for (**e**)) and POm-HO layer 1 axons (***$p = 1 \times 10^{-5}$ for (**f**)), respectively. **g** Distributions of Pearson correlation coefficient between reaction times and calcium response latencies for all axons with significant responses in lick trials ($n = 169$ for VPM-FO, $n = 264$ for POm-FO, $n = 281$ for POm-HO layer 1 axons). Colored bars: Pearson coefficient with $p < 0.05$. **h** Distributions of whisker selectivity index absolute values comparing all axonal populations and lick/no-lick conditions for axons with significant sensory responses in no-lick condition (Kruskal–Wallis two-sided test with Bonferroni correction; VPM-FO: ***$p = 1 \times 10^{-12}$ for lick vs. no-lick, POm-FO: $p = 0.32$ for lick vs. no-lick N.S. not significant, POm-HO: ***$p = 3 \times 10^{-10}$ for lick vs. no-lick, *$p = 0.034$ for VPM-FO vs. POm-FO in lick condition, ***$p = 5 \times 10^{-15}$ for POm-FO and POm-HO in lick condition). Boxplot statistics as in (**a**).

cortical circuits that transfer tactile information to the primary and secondary whisker somatosensory cortex respectively. On the one hand, the lemniscal pathway receives whisker-related activity in Pr5 that is further sent to VPM where neurons project axons mainly in cortical layer 4 of wS1. On the other hand, the

paralemniscal pathway receives inputs in Sp5i that are relayed to POm-FO subdivision where neurons project their axons mainly in cortical layer 4 of wS2.

The Sp5i to POm-FO trigemino–thalamo–cortical circuit that we characterized resembles the previously described extralemniscal

pathway of the rat in terms of axonal innervation in the cortex[2,37] and whisker-related representation[3,7]. The extralemniscal pathway arises from a caudal part of Sp5i projecting to the ventrolateral part of VPM (VPMvl), which in turn innervates layer 4 of wS2 and septal layer 4 domains of wS1. In our experiments, we did not distinguish between rostral and caudal subdivisions of Sp5i. However, in additional anatomical experiments, injections were targeted specifically to the rostral subdivision of Sp5i, which resulted in a similar innervation pattern of POm-FO further projecting in wS2 L4. Further work is needed to investigate functional and anatomical neural circuit differences of caudal and rostral subdivisions of Sp5i in mice. That the domains of POm and VPMvl[2] receiving direct axonal inputs from Sp5i seem to merge in the caudal part of the thalamus[37] may help to understand the organization of these pathways. In our anatomical characterization of the POm-FO pathway, we also sometimes found expression in neurons located in the outer periphery of the VPM, likely corresponding to VPMvl (Supplementary Fig. 1), potentially suggesting that POm-FO and VPMvl could be part of a common circuit. Our data thus suggest two major trigemino–thalamo–cortical pathways conveying parallel tactile information from the periphery to the thalamorecipient layer 4 of wS1 and wS2 with first-order synaptic properties[38], which challenges the classical hierarchal model of the whisker sensory system and suggests that wS2 can process sensory information independently from wS1[39].

The third nucleus that we could isolate is the complementary subdivision of POm, presumably receiving inputs only from the cortex as suggested in previous work[17–19]. Indeed, contrary to the classical view, these POm-HO neurons do not appear to receive tactile inputs from the periphery (Fig. 1c), which could explain their functional responses displaying a lack of whisker-selectivity. Thalamocortical axons from these neurons are located predominantly in layer 1 and layer 5A, and are of the matrix type spanning large regions of the cortex horizontally[40]. Neurons in POm-HO are driven by the layer 5 and layer 6 of the cortex[17,18,28] and could thus serve as a hub for cortico–thalamo–cortical communication, for example linking activity in wS1 and wS2[41].

It has been hypothesized that different types of thalamocortical inputs, core versus matrix, might play different roles based on their anatomical organization. Core-type axons are confined to small cortical domains and provide highly specific sensory information whereas matrix-type axons span large horizontal domains of the cortex and seem to provide broadly tuned sensory information[40]. A possible role of HO thalamic inputs could reside in their ability to alter the functional connectivity of large cortical networks allowing different cortical areas involved in complementary computations to communicate under certain conditions[42]. In view of recent results suggesting a prominent role of layer 1 POm thalamocortical signals in coupling synaptic inputs in distal dendrites to somatic activity in L5 pyramidal neurons[33], we hypothesize that the untuned whisker-evoked responses we observed in POm-HO axons could contribute to facilitate sensorimotor integration in a context-dependent manner. The contribution of POm-HO could therefore be complementary to VPM-FO and POm-FO that provide specific sensory information to specialized distinct cortical domains.

We found that VPM-FO, POm-FO, and POm-HO differ significantly in terms of their whisker selectivity with VPM-FO displaying sharp selectivity, POm-HO displaying little whisker selectivity and POm-FO displaying mixed selectivity. This observation is in line with the receptive field properties of cortical neurons that are the target of these thalamocortical projections. Indeed, many neurons in the wS1 are known to have sharp whisker selectivity[43–45] much like VPM-FO axons, whereas neurons in wS2 typically respond to whisker stimulation with larger receptive fields[39,46]. Similarly neurons in superficial layers of wS1 with large receptive fields were shown to receive inputs from POm thalamic neurons[47]. In terms of response latency, we found that VPM-FO axons responded faster than POm-FO and POm-HO. Although calcium signals offer a lower temporal resolution than electrophysiological recordings, this difference is in line with previous reports[7,28]. Interestingly POm-FO appeared to be composed of two populations, either with short latencies and strong whisker selectivity or with longer latencies and reduced whisker selectivity. This could reflect different cells in POm-FO that are predominantly driven by cortical inputs or trigeminal inputs[19].

In wS1, the spatial organization of VPM-FO axon activity was consistent with whisker specific barrels. In contrast, the somatosensory map of wS2 is less clearly defined than the one observed in wS1[48], and we found spatially intermingled thalamic axons with different whisker-selectivities. However, in our experiments we only imaged small fields of view, and we only stimulated neighboring whiskers, and thus our results do not exclude that POm-FO axons are spatially functionally organized somatotopically at a larger scale.

The VPM-FO and POm-FO pathways also differ in their sensory representation during goal-directed behaviors with the former responding more strongly to self-initiated movements and the latter responding preferably to externally triggered unpredictable whisker movements[49,50]. Distinct sensory representations of self-initiated and external sensory information in parallel thalamocortical circuits have also been reported in the mouse visual system[51] and could be a general feature of sensory systems. Moreover, this result suggests that POm-FO axons conserve their selectivity to whisker stimulation regardless of behavioral conditions, therefore appearing to relay somatosensation while suppressing sensory information resulting from self-initiated movements. POm is targeted by GABAergic neurons in the zona incerta which exert a strong inhibitory influence on the responses of POm neurons[19,52]. Neurons in zona incerta are also under the control of cortical neurons in the whisker primary motor cortex[53] that could provide a top-down signal responsible for shaping the response properties of POm neurons during behavior.

It is interesting to note that all three thalamic populations responded strongly to whisker stimuli. Previous studies in anesthetized rodents[18,19,52], have reported weak sensory responses in POm neurons. However, POm neurons are markedly more active in awake[33,49], alert[32] and active[31] states, perhaps because of state-dependent suppression of inhibitory zona incerta neurons innervating POm[19,53]. Disinhibition of POm during awake states likely allows sensory-evoked responses, as observed in our study, without the need of inactivating zona incerta, as previously reported under anesthesia[19,52].

When the whisker-evoked responses of thalamic axons was characterized during goal-directed sensorimotor tasks, we found that VPM-FO axonal responses were not modulated by lick/no-lick decisions during an early time window following stimulus presentation in line with a recent report[34]. In contrast POm-FO and POm-HO inputs displayed decision-related response modulation early after whisker stimulation, which might play a role in perceptual decision-making beyond the classical sensory relay model[54]. Signatures of decision-related signals have been previously reported bidirectionally between wS1 and wS2[34,55–57], yet the origin of this signal remains unknown. Our results indicate that a decision-related signal can emerge as early as in the thalamus and could therefore be part of a closed-loop circuit designed to maintain important perceptual information through recurrent excitation of cortex and thalamus[58]. The specific role of POm-FO axons in conveying tactile signals to wS2 that are amplified during decision-making is consistent with the

hypothesis that wS2 is a key node in the brain network involved in whisker-based perceptual decision-making[34,55–57,59].

In future experiments, it will be of great interest to investigate the neuronal circuit mechanisms, including possible roles for top-down input from various cortical regions, contributing to decision-making related activity in POm and its relative insensitivity to self-generated input. Future experiments involving optogenetic perturbation of specific thalamic populations during simultaneous functional imaging of other thalamic pathways in awake mice performing tactile tasks will be essential to examine causal neural circuit mechanisms. One interesting approach might be to inject Cre and Flp expressing anterograde trans-synaptic viruses in Pr5 and Sp5 to be able to express different combinations of fluorophores, optogenetic actuators, and activity reporters for more detailed analyses of specializations, correlations, and interactions.

## Methods

**Animals, viral vector injections, and headplate implantation.** Experiments were carried out in mice under protocols approved by the Swiss Federal Veterinary Office (License number VD1628) and were conducted in accordance with the Swiss guidelines for the use of research animals. C57BL/6 wild-type mice and hetero-zygote Gpr26-Cre mice[60] (Tg(Gpr26-cre)KO250Gsat, JAX mouse number 4847098) were housed in cages containing 1–5 mice under a 12/12-h reverse light cycle. The ambient temperature in the animal facility was 23 °C and the relative humidity was maintained around 50%. For all experiments, we used adult mice from both sexes and aged between P25 and P300. For viral injections, mice were first deeply anesthetized with 4% isoflurane mixed in oxygen. They were then placed in a stereotaxic surgery frame using a mouth clamp. Carprofen was injected intraperitoneally as an analgesic (100 μl at 0.5 mg ml$^{-1}$ or 100 μl at 1.5 mg ml$^{-1}$). After repeatedly disinfecting the top of the mouse head with liquid betadine and ethanol 70%, a mixture of lidocaine and bupivacaine was injected under the scalp for local anesthesia. A heating blanket with a closed-loop temperature control system was used to maintain body temperature at ~37 °C. Throughout the surgery, temperature and breathing were monitored closely. Eyes were covered with oint-ment (Viscotears, Alcon, USA; VITA-POS, Pharma Medica AG, Switzerland) to prevent drying. A small scissor was used to open the scalp and expose the skull. Cotton swabs and a scalpel were then used to clean the skull and remove remaining tissue. Lateral muscles on the skull were detached using a scalpel to access the somatosensory cortices. After careful alignment of the skull on the stereotaxic frame, we identified the region of interest for injections. For all targeted subcortical structures, coordinates were measured from bregma as follows: Pr5 (5 mm posterior, 1.8 mm lateral, 3.5 mm deep from bregma), Sp5i (6.5 mm posterior, 1.8 mm lateral, 4.1 mm deep from bregma), VPM (1.7 mm posterior, 1.8 mm lateral, 3.25 mm deep from bregma), POm (2 mm posterior, 1.25 mm lateral, 3.1 mm deep from bregma). A small craniotomy of about 0.5 mm diameter was made at the targeted location and forceps were used to lift the bone cap to access the brain. A thin glass pipette (PCR Micropipets 1—10 μl, Drummond Scientific Company, USA) was first pulled and then the tip was broken using a tissue to give a 21–27 μm inner tip diameter. The pipette was filled with mineral oil and then tip-filled with the AAV vector. The pipette was lowered to the location in the brain very slowly and injection was performed using a single-axis oil hydraulic micromanipulator (Narishige, Japan).

To express Cre-recombinase in the trigeminal nuclei and in the thalamus through anterograde trans-synaptic transfection[21], we used the viral vector AAV1.CamKII0.4. Cre.SV40 (UPenn Vector Core, AV-1-PV2396) and delivered it to the brainstem ipsilateral to the whiskers of interest (right whiskerpad). For injections in the thalamus, several Cre-dependent viral vectors were used: AAV9.CAG.FLEX.tdTomato (UPenn Vector Core, AV-1-ALL864), AAV5.EF1a.DIO.hChR2(H134R)-EYFP-WPRE-HGH (Addgene, 20298-AAV5), AAV1.hSyn.DFO.ChR2-EYFP (Addgene plasmid 136916, virus from Prof. Yizhar, Weizmann Institute of Science, Israel), AAV1.Syn.FLEX. GCaMP6s.WPRE.SV40 (UPenn Vector Core, AV-1-PV2821). Injections were done at an approximate rate of 100 nl min$^{-1}$. For all injections in the brainstem, we used two depths 300 microns apart and injected 250 nl at each depth. For all thalamic injections, we used the same strategy but injected half the amount of viral vector in the hemisphere contralateral to the whiskers of interest. For Supplementary Fig. 2, we followed the same procedure and injected 100 nl of Cholera Toxin subunit B (CTB) conjugated with Alexa647 (Life Technologies, USA) in the posterior part of POm (2.3 mm posterior, 1.4 mm lateral, 3 mm deep from bregma). After all injections, we waited ~5 min before slowly retracting the pipette from the brain. A drop of Kwik-Cast sealant (World Precision Instruments, USA) was then applied with a syringe tip on the craniotomy to protect the brain. A custom-made headplate was lowered on the skull and several layers of super glue (Loctite super glue 401, Henkel, Germany) were applied to attach the headplate. We then used self-curing denture acrylic (Paladur, Kulzer, Germany; Ortho-Jet, LANG, USA) to further secure the headplate and create a

well structure around the skull. Once the super glue and denture acrylic dried, we returned the mouse to its home cage and monitored its recovery from anesthesia. For three days following surgeries, we supplied water with approximately 0.2 mg ml$^{-1}$ of Ibuprofen (Algifor Dolo Junior, VERFORA SA, Switzerland) and closely inspected and weighed each mouse daily to ensure good recovery.

**Cranial window surgery for two-photon imaging.** In isoflurane-anesthetized mice implanted with a headplate, we first trimmed their whiskers to keep only the whiskers C2 and B2. Whiskers on the other side of the face were left intact or slightly shortened for convenience during the surgery. Mice were then head-fixed on a platform with a heating pad to keep their body temperature around 37 °C. Eye ointment was applied on their eyes to prevent drying. Intrinsic optical signal imaging was then acquired using repeated whisker stimulations to visualize the intrinsic signal through the skull covered with super glue[35]. Whiskers were inserted in capillary tubes attached to a piezoelectric actuator that produced continuous 10 Hz pulsatile movements for 4 s proceeded by 4 s with no stimuli. This was repeated for at least 10 trials with a 10-s interstimulus interval. Maps were then averaged and compared between the stimulus and quiet windows. Throughout the imaging isoflurane was kept around 1% to obtain strong intrinsic responses in somatosensory cortices. The functional maps were then obtained to locate the region of the whisker primary and secondary somatosensory cortex responding to C2 and B2 whisker stimulation. Mice were then moved back to a surgery table. A circular craniotomy with ~3 mm diameter was then performed over the region of interest. Once the bone cap was removed, we used a custom-made perfusion–suction system to continuously rinse the exposed brain region with Ringer solution. A needle tip was shaped into a hook and used to cut and remove the dura over the whole craniotomy. A piece of razor blade (Wilkinson Sword, UK) was cut to the dimensions of the microprism edge (1.25 mm) and subsequently glued to an injection plunger. Using a micromanipulator we descended the razor blade posterior to the region of interest perpendicular to the cortex and with the blade along the medio-lateral axis. Once at the surface of the cortex, the razor blade was slowly lowered roughly 800 microns into the cortex using a micromanipulator (Luigs and Neumann, Germany) to monitor the depth of the penetration. We next retracted the blade while continuously cleaning the surface of the cortex with Ringer. We then used a custom-made microprism window assembly consisting of two co-aligned 3 mm coverslips on top of a 5 mm coverslip with a microprism (Tower Optical Corporation, USA) glued in the center of the 3 mm coverslip. All optical elements were glued together using a UV-curing optical adhesive (NOA61, Thorlabs, USA). For imaging in the whisker secondary somatosensory cortex, we glued the microprism off-center to access this more lateral region of the cortex. This microprism window assembly was held by a syringe with a flat tip needle attached to a Venturi suction pump and lowered into the craniotomy using the micromanipulator. Great care was taken to penetrate the microprism edge in the incision made with the razor blade. The face of the microprism was oriented toward the anterior part of the cortex. Kwik-Cast sealant or UV-Curing Optical Adhesives (NOA61, Thorlabs, USA) was used to isolate the edge of the craniotomy around the window. Finally, super glue and self-curing denture acrylic were applied around the edge of the 5 mm coverslip to maintain the cranial window firmly in place.

**Two-photon calcium imaging.** Axonal imaging was performed using a custom-made two-photon microscope[27]. The microscope was equipped with a galvo-resonance mirror pair (8 kHz CRS, Cambridge Technology, USA), allowing a frame rate of 30 Hz for resolutions of either 512 × 512 pixels or 512 × 1024 pixels with the frame length being along the resonant scanner axis. A femtosecond tunable infrared laser line (Mai Tai, Spectra Physics—Newport, USA) was fed into the light path at a wavelength of 940 nm to excite the genetically encoded calcium indicator GCaMP6s[61]. Light emission was detected with a GaAsP photosensor module (H10770PA-40, Hamamatsu, Japan), and signal acquisition was performed with National Instrument hardware (NI PXIe-1073, NI PXIe-6341, National Instruments, USA). The microscope head was movable and controlled in three dimensions by motors (Luigs and Neumann, Germany). A 16× immersion objective (16× Nikon CFI LWD, Japan) was used for all the imaging. The system was operated by the Matlab-based software ScanImage SI5 (Vidrio Technologies, USA). To image thalamic axons, we used a 3× numerical zoom in ScanImage. For each mouse, multiple imaging sessions were performed at very different depths and locations within the field of view in the microprism. During the acquisition, we used a trial-based acquisition scheme where acquisition sequences of fixed duration (9 s) were triggered at the beginning of each trial with intertrial intervals where no acquisition was performed.

**Perfusion and postmortem analysis.** Mice were anesthetized with isoflurane and overdosed with pentobarbital. They were then perfused with 4% paraformaldehyde (PFA), and their brains were removed. Brains remained in 4% PFA overnight, then transferred into PBS for two days. Next, 100 μm coronal sections were cut on a vibratome (Leica VT1200S). In some cases, we amplified the eYFP signal with immunostaining. To do so, the slices were firstly incubated in a blocking buffer containing 0.3% Triton X-100 (Applichem, Germany) and 2% normal goat serum

(NGS, Vector, S-1000-L020) in PBS (0.9% NaCl, 0.01 M phosphate buffer, pH 7.4) for 1 h. Then, we incubated with primary anti-GFP antibody (rabbit polyclonal 1:5000, Abcam 290, UK) together with 0.3% Triton X-100 in PBS for 48 h shaking at 4 °C followed by two washes with PBS for 10 min. Subsequently, the slices were incubated for 2–2.5 h at room temperature with the secondary antibody (goat anti-rabbit conjugated to Alexa 488 1:200, Life Technologies A-11012) together with 0.3% Triton X-100 in PBS. Afterwards, we washed the slices with PBS 3 times for 10 min and in the second wash we added DAPI (25 µl of DAPI in 10 ml PBS) in order to stain the cell nuclei. Finally, the slices were mounted on Superfrost slides using 1,4-Diazabicyclo[2.2.2]octane (DABCO, Sigma-Aldrich D27802, USA). Images were obtained using an epifluorescence microscope (Olympus Slide Scanner VS120-L100 or LEICA DM 5500) through a 10×/0.40 NA air objective.

**Two-photon serial-section tomography**. Some brains were imaged through two-photon serial-section tomography. After post-fixation, the brains were embedded in 5% oxidized agarose (Type-I agarose, Merck KGaA, Germany) and covalently crossed-linked to the agarose by incubating overnight at 4 °C in 0.5–1% sodium borohydride (NaBH₄, Merck KGaA, Germany) in 0.05 M sodium borate buffer. Then, we imaged the brains using a custom-made two-photon serial-section microscope, which was controlled using the MATLAB-based software ScanImage 2017b (Vidrio Technologies, USA) and BakingTray (https:// github.com/BaselLaserMouse/BakingTray, extension for serial sectioning)[62]. The imaging setup consisted of a two-photon microscope coupled with a vibratome (VT1000S, Leica, Germany) and a high-accuracy X/Y/Z stage (X/Y: V-580; Z: L-310, Physik Instrumente, Germany). The thickness of physical slices was set to be 50 µm for the entire brain and we acquired optical sections at 5 µm using a high-precision piezo objective scanner (PIFOC P-725, Physik Instrumente, Germany) in two channels (green channel: 500–550 nm, ET525/50, Chroma, USA; red channel: 580–630 nm, ET605/70, Chroma, USA). Each brain section was imaged with 7% overlapping 1025 × 1025 µm tiles. We used a 16× water immersion objective lens (LWD 16×/ 0.80 W; MRP07220, Nikon, Japan), with a resolution of 0.8 µm per pixel in X and Y and measured axial point spread function of 5 µm full width at half maximum. After acquisition, the raw tiles were stitched using the MATLAB-based software StitchIt (https://github.com/SainsburyWellcomeCentre/StitchIt). This software applies illumination correction based on the average tile in each channel and optical plane and subsequently stitches tiles for the entire brain. After stitching and before further image processing, we down-sampled the stitched images by a factor of 6 in X and Y obtaining a voxel size of 4.8 × 4.8 × 5 µm, using the MATLAB-based software MaSIV (https://github.com/SainsburyWellcomeCentre/masiv).

**Whisker stimulation and behavioral training**. Mice were trained in a two-whisker discrimination task under a water restriction schedule. Two piezoelectric actuators were mounted in a two-arm holding system with foam to dampen vibration resonance. Small capillary tubes were glued to the piezoelectric element to insert each whisker. The tip of each tube was melted to slightly close the opening so that each whisker was tightly held inside the tube with no free space for movement. A spout was presented within the reach of the tongue. A piezo-film was attached to the spout in order to detect licking activity as vibrations. Water reward was delivered through the spout using a valve system. Facial movements were filmed at 100 frames per second using a high-speed camera (CL 600 ×2/M, Optronis, Germany) and an infrared light outside the visible range of the mouse. Sensory stimulation and behavioral training were performed through a Matlab-based (Mathworks, USA) custom-made software. Whisker stimuli consisted of 5 sine waveform pulses, each lasting 40 ms for a total stimulus duration of 200 ms. The amplitude of the tube displacement was ~1 mm and was comparable for both whiskers. This value was well beyond the detection threshold. We used trials with a duration of 9 s. Three possible stimulus conditions were considered: stimulation of the C2 whisker, stimulation of the B2 whisker or no stimulation at all. Each trial started with a quiet window of 2 s during which lick detection would result in canceling the trial and starting over. At the end of the quiet window, stimuli were delivered at the start of a 2 s long response window during which mouse could lick the spout. If a lick was detected upon C2 whisker stimulation, a water reward of ~8 µl was delivered. Licking upon B2 whisker stimulation resulted in a 10 s timeout, and licking in absence of whisker stimuli had no effect. At least 4.5 s separated each trial resulting in a minimum interstimulus interval of 13.5 s. During the first session of the training, mice were head-fixed and stimulation of the C2 whisker was automatically accompanied by water reward, whereas B2 whisker stimulation was delivered without reward. This associative phase could be stopped during the first session if the mouse started licking the spout in response to whisker stimulation and did not last more than one session. After a few days, on average, mice started performing the task with performance above chance level (percentage of correct trials larger than 60%). Analysis of axonal calcium data only included these sessions. Trials with whisker stimuli constituted 80% of all trials and no stimulus trials 20%. Detection of licking was done in two different ways. During the behavioral training, licking was detected through strong vibrations of the spout. After the session, the movie of facial movements was analyzed using a dimensionality reduction algorithm and t-distributed stochastic neighbor embedding (t-SNE)[63] to classify movements[64]. After applying principal component analysis decomposition on individual frames, we kept the 50 principal components and ran wavelet decompositions over the temporal domain with 25 frequency bands. t-SNE was then applied to the resulting reduced space to obtain two-dimensional maps of orofacial dynamics. Lick reaction times were extracted from this analysis by finding the first time bin where lick movement could be identified. This resulted in reaction times smaller than the ones obtained with spout contact because they represent the initiation of movement instead of the timing when the tongue is protracted.

**Whole-cell brain slice recordings**. The brains of adult mice of either sex were removed 3–4 weeks after viral injections and 300-µm-thick parasagittal (35° away from vertical) brain slices were cut on a vibrating slicer (Leica VT1200S, Germany) in an ice-cold modified artificial cerebrospinal fluid (ACSF) containing (in mM) 87 NaCl, 25 NaHCO₃, 25 D-glucose, 2.5 KCl, 1.25 NaH₂PO₄, 0.5 CaCl₂, 7 MgCl₂, 75 sucrose, aerated with 95% O₂ + 5% CO₂. After being sliced, the tissue was transferred to a chamber with the same solution at room temperature for 25 min. Then, the tissue was transferred to a chamber with standard ACSF, containing (in mM) 125 NaCl, 25 NaHCO₃, 25 D-glucose, 2.5 KCl, 1.25 NaH₂PO₄, 2 CaCl₂, 1 MgCl₂, aerated with 95% O₂ + 5% CO₂ at room temperature. Slices were maintained at room temperature until the recording session started (within 3 h of slicing). The brain slices containing ChR2-eYFP expressing axons were identified with a 4× objective lens (Olympus UPlanFI 4×, 0.13 NA) using very brief illumination with 470-nm light to excite eYFP fluorescence. Creation of a gradient contrast image of cells was achieved by transmitted light through a Dodt contrast element (Luigs and Neumann, Germany). Brain slices were continually superfused with ACSF containing 50 µM picrotoxin (PTX), 1 µM tetrodotoxin (TTX), 100 µM 4-aminopyridine (4-AP) at 34 °C and aerated with 95% O₂ + 5% CO₂ mixture[24]. The membrane potentials of neurons were recorded in whole-cell configuration with a Multiclamp 700B amplifier (Molecular Devices, USA). Borosilicate patch pipettes with a resistance of 5–7 MΩ were used. The pipette intracellular solution contained (in mM) 135 K-gluconate, 4 KCl, 4 Mg-ATP, 10 Na₂-phosphocreatine, 0.3 Na-GTP, and 10 HEPES (pH 7.3, 280 mOsmol l⁻¹). Biocytin (3 mg ml⁻¹) was added to the intracellular solution. Electrophysiological membrane potential data were low-pass Bessel filtered at 10 kHz and digitized at 20 kHz with an ITC-18 acquisition board (Instrutech, USA). Data acquisition routines were custom-made procedures written in IgorPro software (Wavemetrics, USA). Membrane potential measurements were not corrected for the liquid junction potential. For stimulation of ChR2-expressing axons, we used a 470-nm collimated blue LED system (Thorlabs, USA) coupled to a 1 mm optic fiber (Thorlabs; 0.48NA, USA). Optic fiber blue light stimulation had a peak light power of ~30 mW at the tip of the fiber. Light power varied across experiments between ~1 and ~30 mW. After completion of the electrophysiological recordings, slices were fixed for at least 24 hours in 4% paraformaldehyde and then transferred into phosphate-buffered saline (PBS). Slices were then washed in PBS three times over a period of 1 hour. After washing, slices were then incubated in a blocking solution containing 5% normal goat serum (NGS) and 0.3% Triton X-100 for 1 h. Then slices were transferred into the staining solution containing 0.3% Triton X-100 and 1:2000 of Streptavidin conjugated to Alexa 647 (Life Technologies, USA). Slices were incubated for 3 h and then washed with PBS at room temperature. DAPI was used as a counterstain. Slices were then mounted and imaged under a confocal microscope (Leica SP8 FLIM, Germany) through a 25×/0.95NA water objective (HC Fluotar). All the recovered neurons could be identified and matched to the electrophysiological recording.

**Data processing**. Two-photon calcium signals from thalamic axons were extracted from imaging sessions using the Matlab-based toolbox Suite2p[65]. After correcting each imaging session for rigid motions, the toolbox identified regions of interest (ROI) corresponding to axonal segments. The time-varying signal was extracted from these ROIs. Axonal segments belonging to the same cell could be present in different parts of the image with no direct connections between them. Therefore we used a correlation-based clustering analysis to identify sets of axonal segments sharing strong correlation (>0.8). Signals from highly correlated axonal segments were averaged together weighted by the number of pixels for each ROI. We then inspected the resulting ROIs and the level of correlation between axonal segments. If additional stretches needed to be merged, we use a graphical user interface on Matlab (MathWorks, USA) to perform this stitching operation with manual inspection. Once ROIs had been defined, the neuropil signal was subtracted using the same pixel-based weights. The neuropil signals were extracted from the Suite2p algorithm together with the corresponding estimated coefficients. The resulting calcium signal was then normalized to the noise level. We first estimated the signal mode value in a piecewise manner in segments of 3000 frames. This baseline estimation was then fitted by a fifth order polynomial function to filter out spurious high-frequency components. This baseline was subtracted from the ROI signal. Because of the nonnegative nature of GCaMP6s signal fluctuations, the noise level was estimated through the distribution of negative fluctuations below the baseline. Assuming a Gaussian noise model, we divided the baseline-subtracted calcium signal by the standard deviation of this Gaussian noise to obtain traces normalized to noise level. When considering stimulus-evoked responses we computed a z-score relative to baseline GCaMP6s activity. To do so we collected signals in the quiet window 0.5 s prior to the response window. The mean of this distribution was

subtracted from the signal, which was further divided by the standard deviation of the distribution.

**Data analysis**. The z-score signal obtained for each ROI could then be used to perform one-sided z-tests on each time bin over trials of the same condition. Doing so we obtained a p value for each time bin. The time-varying p values were then used to assess significance for each axon and condition. For a response to be significant, the function $-\log_{10}(p\ value)$ should exceed the value ten in at least three consecutive bins. This criterion, based on the slow dynamics of GCaMP6s, was used to avoid spurious significance. For responses that passed the significance test, the response latency was computed by finding the first bin where the function $-\log_{10}(p\ value)$ exceeded the value 5. Axons displayed whisker sensory responses if the significance criterion was met during C2 and/or B2 whisker stimulations in the absence of licking. Axons that did not respond significantly in these conditions could show significant responses during lick events in the absence of whisker stimuli or when both are present. Whisker selectivity indices were measured by first averaging responses to C2 and B2 whiskers over the response window (2 s following stimulus onset). The index was computed as the normalized difference between the two: $(C2 - B2)/(C2 + B2)$. We quantified the distribution of VPM-FO and POm-FO highly tuned axons by separating all pixels within a field of view that show strong tuning to either B2 (WSI < −0.75) or C2 (WSI > 0.75). We then used the correlation matrix of each pixel distribution to compute a confidence interval with $p = 90\%$ defining ellipses containing pixels responding to either whiskers. We define an overlap index as the surface of the intersection of these ellipses normalized by the average ellipse size between the two populations. This index is equal to 0 if the two ellipses are not intersecting and is equal to 1 if the two domains are identical. To analyze the early calcium response in lick trials we chose a window that was shorter than the average reaction time (0.29 s) but long enough to capture the calcium response elicited by whisker stimulation given that GCaMP6s has a slow rise time. We decided to use 0.266 s to have a long enough window below the typical reaction time. Since our acquisition frame rate with the two-photon microscope (galvo-resonant mirror pair) was set around 30 frames per sec (corresponding to 0.033 s per frame), 0.266 was the last frame before the average reaction time. The result of this analysis is robust with different windows as we found qualitatively similar results with slightly larger (0.4 s) or smaller (0.2 s) windows. To compute the timing correlation between lick reaction time and calcium responses, we used lick trials where the B2 whisker was stimulated, because the licking pattern for non-rewarded trials was similar across trials. If a response to passive whisker stimulation is present, the first component is locked to the onset of this stimulation. It is therefore assumed to have a fixed latency across lick trials. If, in addition, calcium responses were evoked by lick-related events then a second component is present in the calcium response for which the latency should vary with first lick reaction time. In our analysis we try to capture the dominant calcium response component and identify whether it is locked on lick reaction time. Therefore for each trial we computed the cumulative distribution of the signal starting at the onset of the response window and for a 3 s-long time window. Some axons displayed a first response component to the B2 whisker stimulation followed by a second stronger lick-related response. The latency of the former was invariant across trials whereas the timing of the second could be correlated to the reaction time. To prevent B2 whisker sensory responses from biasing the response latency correlated with licking, we identified the 50% percentile time bin over the cumulative distribution and used it as latency. The timing correlation was then computed with the Pearson coefficient between lick reaction time and response latency.

**Statistical tests**. In order to assess the significance of our results we used paired Wilcoxon signed-rank tests when the same axonal populations were compared between two conditions (lick compared to no-lick), unpaired Kruskal–Wallis test when populations of different sizes were compared. All tests used in this paper were two-sided except for z-test performed over trials. No blinding or randomization of samples was done in any of our analyses. Variances were computed for all groups and were generally in the same order of magnitude.

**Reporting summary**. Further information on research design is available in the Nature Research Reporting Summary linked to this article.

## Data availability
The data used to generate figures that support the findings of this study are freely available in the Open Access CERN database Zenodo: https://zenodo.org/communities/petersen-lab-data with doi hyperlink: https://doi.org/10.5281/zenodo.3824359.

## Code availability
The Matlab code used to generate figures that support the findings of this study are freely available in the Open Access CERN database Zenodo: https://zenodo.org/communities/petersen-lab-data with doi hyperlink: https://doi.org/10.5281/zenodo.3824359.

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

## Acknowledgements

We thank Samuel Gex and the EPFL Faculty of Life Sciences workshop for technical assistance in the building of the two-photon microscope and the whisker stimulator system. We thank Johannes Mayrhofer, Denis Jabaudon and Anthony Holtmaat for insightful discussions and comments on the manuscript. We thank V. Jayaraman, R.A. Kerr, D.S. Kim, L.L. Looger, and K. Svoboda from the GENIE Project, Janelia Farm Research Campus, Howard Hughes Medical Institute (HHMI), for the distribution of GCaMP6. This work was supported by the Swiss National Science Foundation (310030B_166595, 31003A_182010 and CRSII5_177237), the European Research Council (ERC-2011-ADG 293660) and Marie Skłodowska-Curie Fellowship FP7-PEOPLE-2010-IOF (274920).

## Author contributions

S.E.-B., B.S.S and C.C.H.P. conceived the experiments. S.E.-B. performed the surgeries and viral injections, carried out in vivo experiments, and performed the analysis. B.S.S performed in vitro patch–clamp experiments. B.S.S. and G.F. prepared the anatomical samples and carried out the fluorescence imaging. T.B.O. and O.Y. provided the AAV Cre-OFF viral vectors. S.E.-B. and C.C.H.P. wrote the paper with comments from all the authors.

## Competing interests

The authors declare no competing interests.
