## [Peer Review File · Nature Communications]

Reviewers' Comments:

Reviewer #1:

Remarks to the Author:

The manuscript by El-Boustani et al., titled "Anatomically and functionally distinct thalamocortical inputs to primary and secondary mouse whisker somatosensory cortices" dissects the three thalamocortical pathways that carry distinct sets of sensorimotor information to the primary and secondary whisker somatosensory cortices. Utilizing a serotype specific anterograde trans-synaptic transfection property of AAV, the authors successfully labeled and manipulated the three pathways in order to examine some of the pathway specific properties in a behaviorally-relevant manner. Specifically, the study confirmed the following routes: 1) Pr5 – VPM-FO – wS1:L4, 2) Sp5i – POM-FO – wS2:L4, and 3) wS1&2: L5&6 - POM-HO – wS1&2: L1&5. Behavioral tasks combined with calcium imaging of the thalamocortical axons revealed their differential responses to passive whisker sensation, self-initiated whisker movements, and reward-related decision making. VPM-FO axons responded to passive whisker stimulation with strong whisker-specific preferences which were overridden by self-initiated whisking associated with licking behavior. POM-FO and POM-HO axons displayed enhanced transient response during the early phase of lick trials compared to no lick trials, indicating their potential roles in decision-making. POM-FO axons showed no strong correlation with the lick reaction time and conserved their whisker selectivity regardless of the motor output. The study is of impressively high quality with top notch modern techniques that the authors seem to command with ease. However, some of the findings, namely the reidentification of the lemniscal and paralemniscal pathways fall short on novelty, and others that could be novel and intriguing would benefit from further analyses. The authors do a great job of giving credit where it's due by citing previous publications, but perhaps they should better highlight and articulate the novel part of their findings for a broader audience. Related to this, the current manuscript reads rather observational and lacks in attempts to put their findings in the context of a "big picture". My specific comments are below:

Major comments/discussions:

1. The finding that POM-FO axons are intermingled in the cortex and carry mixed properties is intriguing. What kind of signals do you get from a stretch of axons that pass through both C2 and B2 domains? Does weak vs strong whisker selectivity correlate with how intermingled the axons are? It is also interesting that in this study unlike previous report, there was no need for inhibiting zona incerta to uncover the two subpopulations. Why do you suppose that is?
2. Regarding response properties of the thalamocortical axons (Figure 4):
 - a. VPM-FO axons: Did the offset of the prolonged Ca response of the VPM-FO axons correlate with the cessation of licking?
 - b. POM-FO axons: Despite the finding that there are two distinct subpopulations of POM-FO inputs, those with tuned response and short latency and others with untuned response and longer latency (presumably indicative of those driven by Sp5i driver inputs vs others driven by cortical top-down inputs), these are unfortunately lumped together in the analyses described in Figure 4. If these two subpopulations could be analyzed separately, would the former resemble the properties of VPM-FO axons and the latter those of POM-HO axons?
 - c. POM-HO axons: Why is the whisker selectivity of POM-HO axons, which is already quite low to begin with, further reduced by licking?
3. Question regarding the big picture: How do the three parallel inputs influence each other at the level of cortex? It is mentioned briefly in the discussion that they "might act in concert at the cortical level", but how? According to Figure 1, VPN neurons project to both layer 4 and deep layer 5b/6. Does the whisker sensation information carried by VPN neurons ultimately reach the POM-HO via layer 6 or

layer 5? What would happen to the POM-HO activity during discrimination task if VPN axons in the cortex were inhibited via optogenetics? Or for a simpler test, would optogenetic activation of VPN axons in the cortex induce any response in the POM-HO neurons/axons? This is a big question, and I don't want to be one of those reviewers who ask for impossible additional experiments. But since the authors do already possess the ability to optogenetically manipulate and perform calcium imaging on these thalamocortical inputs, it would be fantastic to see some manipulation experiments (but not required). At the very least, some more discussion on this point would add to the value of the manuscript in my view.

Minor comments:

1. Segregated axons from VPM vs intermingled axons of POM-FO: How did you define/quantify segregated vs intermingled?
2. Fig 4d-f: how was the duration of the grey box (0.266sec) chosen? If I understand it correctly, it is longer than the duration of the whisker stimulation (0.2sec) but shorter than the average reaction time to lick (0.29sec).
3. On page 8, it is stated that the early phase of the VPM axon responses does not differ whether the mouse licks or doesn't lick the spout, but the calcium response latencies do indeed vary with first lick reaction time. These two seemingly contradicting statements are confusing to me. Is this why the cutoff for the "early phase" was set at 0.266sec? If it was set to the average reaction time to lick of 0.29sec, would you see a difference between lick vs no lick trials?

Reviewer #2:

Remarks to the Author:

The paper by El-Boustani et al. describes distinct thalamocortical pathways to the primary and secondary somatosensory cortices in the mouse. They distinguish three pathways of vibrissa information: a VPM-first order pathway (VPM-FO) that projects to S1, a POM-first order pathway (POM-FO) that projects principally to S2, and a POM-higher order (POM-HO) pathway that projects to layers 5a and 1 across S1 and S2.

They also report how these pathways convey vibrissa information in awake behaving mice. The authors used an impressive variety of methods which include viral approaches, optogenetic stimulation combined with in vitro recordings, two-photon imaging with a microprism, and two-photon calcium imaging in the head-restrained mouse performing a go/nogo task upon vibrissa stimulation. Data are well illustrated and well quantified.

Major point

Prior studies in rats (referred to in the paper) have identified three ascending pathways of vibrissa information; a lemniscal pathway (PrV-VPM-S1), a paralemniscal pathway (SpVIrostral-POM-S1,S2), and an extralemniscal pathway (SpVICaudal-VPMvl-S2). It is not clear whether the POM-FO and POM-HO pathways correspond to the paralemniscal and extralemniscal pathways, respectively. The authors injected a very large amount of anteroAAV expressing Cre in the interpolaris nucleus (500 nl), which likely led to Cre expression in the thalamic targets of both the rostral and caudal divisions of the interpolaris nucleus (e.g., POM and VPMvl). This is a crucial point to clarify to avoid creating confusion in this field of research. As the anteroAAV expressing Cre has no reporter, the authors should consider co-injecting this virus with another AAV expressing a fluorescent reporter. Reducing the volume of injections (less than 100 nl) is mandatory to achieve specificity of labeling in the rostral and caudal aspects of the interpolaris nucleus. This is the main weakness of the paper, which questions the validity of the physiological results (though they are superbly illustrated).

Reviewer #3:
None

Reviewer #1 (Remarks to the Author):

The manuscript by El-Boustani et al., titled “Anatomically and functionally distinct thalamocortical inputs to primary and secondary mouse whisker somatosensory cortices” dissects the three thalamocortical pathways that carry distinct sets of sensorimotor information to the primary and secondary whisker somatosensory cortices. Utilizing a serotype specific anterograde trans-synaptic transfection property of AAV, the authors successfully labeled and manipulated the three pathways in order to examine some of the pathway specific properties in a behaviorally-relevant manner. Specifically, the study confirmed the following routes: 1) Pr5 – VPM-FO – wS1:L4, 2) Sp5i – POm-FO – wS2:L4, and 3) wS1&2: L5&6 - POm-HO – wS1&2: L1&5. Behavioral tasks combined with calcium imaging of the thalamocortical axons revealed their differential responses to passive whisker sensation, self-initiated whisker movements, and reward-related decision making. VPM-FO axons responded to passive whisker stimulation with strong whisker-specific preferences which were overridden by self-initiated whisking associated with licking behavior. POm-FO and POm-HO axons displayed enhanced transient response during the early phase of lick trials compared to no lick trials, indicating their potential roles in decision-making. POm-FO axons showed no strong correlation with the lick reaction time and conserved their whisker selectivity regardless of the motor output. The study is of impressively high quality with top notch modern techniques that the authors seem to command with ease. However, some of the findings, namely the reidentification of the lemniscal and paralemniscal pathways fall short on novelty, and others that could be novel and intriguing would benefit from further analyses. The authors do a great job of giving credit where it's due by citing previous publications, but perhaps they should better highlight and articulate the novel part of their findings for a broader audience. Related to this, the current manuscript reads rather observational and lacks in attempts to put their findings in the context of a “big picture”. My specific comments are below:

We thank the reviewer for his/her insightful comments. In the following, we describe changes made in the manuscript to address all raised issues. In this study, we delineate distinct thalamo-cortical pathways that reconcile previous publications pointing toward subdivisions of POm. By doing so we identify and isolate parallel thalamocortical pathways and describe for the first time to our knowledge their inputs to whisker somatosensory cortices in awake behaving mice. We highlighted a striking difference in somatosensation encoding between two trigemino-thalamo-cortical pathways potentially contributing differently during goal-directed behaviors. Moreover, we revealed that thalamic neurons belonging to the classical paralemniscal pathway believed to mediate brainstem inputs to layer 1 and layer 5a might actually be driven by cortico-thalamic inputs exclusively. Following the reviewer's recommendation, we try to discuss these results within a more general framework of somatosensory integration.

Major comments/discussions:

1. The finding that POm-FO axons are intermingled in the cortex and carry mixed properties is intriguing. What kind of signals do you get from a stretch of axons that pass through both C2 and B2 domains? Does weak vs strong whisker selectivity correlate with how intermingled the axons are?

This is a very interesting question that we try to address through further analyses of our data as suggested by the reviewer. However, we could not reach a clear conclusion due to several technical issues. First, we weren't able to identify clear C2 or B2 domains in the secondary somatosensory cortex (wS2) using intrinsic optical imaging. Although a somatosensory map is present in this area it is much less well-defined than in wS1 (Hubatz et al., 2020). Our data indicate that in the area where C2/B2 intrinsic signals are strong, the axons are not organized into distinct domains showing dominance in C2 or B2 tuned axons. Secondly, since axons in wS2 are not confined within barrel-like structures it is hard to quantify the trajectories of single axons, as they can be visible in different parts of the field of view due to their elongated shapes. This is reflected by separated segments of axons displaying strong correlation during imaging that are thus classified as belonging to the same neurons as described in the Methods section. We can't exclude the existence of spatial organization at a larger scale, but at the level of our imaging field of view, we observed a clear distinction between VPM-FO and POm-FO, indicating reduced or lack of spatial segregation between POm-FO axons tuned to different whiskers compared to VPM-FO axons. This is now explained and quantified on page 7 (see minor point 1 below) and better discussed in the manuscript on page 13:

“In wS1, the spatial organization of VPM-FO axon activity was consistent with whisker specific barrels. In contrast, the somatosensory map of wS2 is less clearly defined than the one observed in wS1^{4b}, and we found spatially-intermingled thalamic axons with different whisker-selectivities. However, in our experiments we only imaged small fields of view, and we only stimulated neighbouring whiskers, and thus our results do not exclude that POm-FO axons are spatially functionally organized somatotopically at a larger scale.”

It is also interesting that in this study unlike previous report, there was no need for inhibiting zona incerta to uncover the two subpopulations. Why do you suppose that is?

Most of previous studies of POM neurons have been conducted in anesthetized animals. In these conditions, POM neurons show reduced activity levels (Masri et al., 2008). Further studies have revealed state-dependent activity of POM neurons, with increases in firing rate during active desynchronized brain states (Urbain et al., 2015). A very recent study compared the level of activity in POM between awake and anesthetized states and how it might affect cortical responses showing that under anesthesia POM activity is strongly reduced (see Fig. S6 in Suzuki and Larkum, 2020). Conversely, the activity of the inhibitory neurons of the zona incerta might be reduced during active states, for example during increased cholinergic tone (Trageser et al., 2006) and during whisker motor cortex activity (Urbain & Deschenes, 2007). We therefore think that the strong responses of POM observed in our data, obtained without inhibition of zona incerta, are because our mice are awake and behaving. During such an active state ZI neurons may be inhibited, thus allowing POM neurons to become excited through disinhibition. This is now discussed in the manuscript on page 14:

"It is interesting to note that all three thalamic populations responded strongly to whisker stimuli. Previous studies in anesthetized rodents^{18,19,52}, have reported weak sensory responses in POM neurons. However, POM neurons are markedly more active in awake^{33,49}, alert³² and active³¹ states, perhaps because of state-dependent suppression of inhibitory zona incerta neurons innervating POM^{19,53}. Disinhibition of POM during awake states likely allows sensory-evoked responses, as observed in our study, without the need of inactivating zona incerta, as previously reported under anesthesia^{19,52}."

2. Regarding response properties of the thalamocortical axons (Figure 4):

a. VPM-FO axons: Did the offset of the prolonged Ca response of the VPM-FO axons correlate with the cessation of licking?

We thank the reviewer for suggesting this analysis. Indeed VPM-FO axons show a strong decay of activity following the offset of a licking bout, which is much stronger at the population level than what is observed for POM-FO and POM-HO. Furthermore, we compared responses evoked by single licking events to responses evoked by passive whisker stimulations. We found that VPM-FO axons display responses for these two events of comparable magnitude, whereas POM-FO and POM-HO show a strong preference for passive whisker stimulation, significantly different from VPM-FO axons. This is now summarized in a new Supplementary Figure 4.

b. POM-FO axons: Despite the finding that there are two distinct subpopulations of POM-FO inputs, those with tuned response and short latency and others with untuned response and longer latency (presumably indicative of those driven by Sp5i driver inputs vs others driven by cortical top-down inputs), these are unfortunately lumped together in the analyses described in Figure 4. If these two subpopulations could be analyzed separately, would the former resemble the properties of VPM-FO axons and the latter those of POM-HO axons?

We performed separate analyses for tuned and untuned POM-FO axons and present this analysis in a new Supplementary Figure 5. As anticipated by the reviewer, some properties of highly tuned POM-FO axons resemble those of VPM-FO, although with a more moderate effect size (small reduction of whisker selectivity during licking and weak timing correlation). However, we found that both of these two subpopulations still display strong modulation in the early phase comparing lick and no lick conditions, which was not observed for VPM-FO. This indicates that although Sp5i might dominantly drive tuned POM-FO axons, these axons are still strongly influenced by top-down inputs resulting in response properties distinct from the lemniscal pathway. This is discussed in the manuscript on page 9:

"Subdividing POM-FO axons into highly-tuned and untuned populations revealed some resemblance of the tuned axons with VPM-FO axons although influence of licking on response properties was much weaker and both tuned and untuned axons displayed a strong modulation in the early response phase (Supplementary Fig. 5), indicating an important contribution of top-down inputs not seen in VPM-FO axons."

c. POM-HO axons: Why is the whisker selectivity of POM-HO axons, which is already quite low to begin with, further reduced by licking?

We thank the reviewer for pointing this out. To illustrate this phenomenon more clearly we added examples of POM-HO axonal responses during the goal-directed behavior that were not included in the original manuscript (see new Supplementary Figure 3). Indeed we observed that some axons that were weakly tuned to a whisker would completely lose this selectivity as soon as the animal licks in response to whisker stimulation. Interestingly these responses during lick trials were amplified prior to the first lick reaction time, as shown in the examples with no correlation to licking timing. Although we haven't identified the origin of this functional change, we hypothesize that they could result from the overall increase in cortico-thalamic feedback during lick trials. This modulation might be favorable for enabling dendritic integration of top-down, including motor, inputs in cortical pyramidal neurons in wS1 and wS2 as recently suggested (Suzuki and Larkum, 2020). This is now mentioned in the manuscript and discussed in the caption of the new Supplementary Figure 3.

3. Question regarding the big picture: How do the three parallel inputs influence each other at the level of cortex? It is mentioned briefly in the discussion that they “might act in concert at the cortical level”, but how? According to Figure 1, VPN neurons project to both layer 4 and deep layer 5b/6. Does the whisker sensation information carried by VPN neurons ultimately reach the POM-HO via layer 6 or layer 5? What would happen to the POM-HO activity during discrimination task if VPN axons in the cortex were inhibited via optogenetics? Or for a simpler test, would optogenetic activation of VPN axons in the cortex induce any response in the POM-HO neurons/axons? This is a big question, and I don’t want to be one of those reviewers who ask for impossible additional experiments. But since the authors do already possess the ability to optogenetically manipulate and perform calcium imaging on these thalamocortical inputs, it would be fantastic to see some manipulation experiments (but not required). At the very least, some more discussion on this point would add to the value of the manuscript in my view.

The experiments suggested by the reviewer are very interesting but we think they are beyond the scope of this study as they present some important challenges. Studying the contribution of VPM-FO inputs to the responses of POM-HO requires experiments that separately target these two populations with a depolarizing or hyperpolarizing opsin (VPM-FO) and GCaMP6s (POM-HO), respectively. This is the first major difficulty of these experiments as we use the viral vector AAV1.CaMKIIa.Cre to specifically target each of these populations, and in order to independently target the distinct signaling pathways, we would need to develop appropriate orthogonal viral vectors, perhaps based around Flp recombinase (as now explicitly discussed in the manuscript). The other technical challenge has to do with the combination of two-photon calcium imaging with GaAsP photomultiplier tubes (PMTs) and optogenetic stimulation. Imaging can’t be performed or is interrupted while the optogenetic light is on, which would prevent us from tracking the response dynamics of POM-HO axons following or during optostimulation. This could be prevented using a fast-shutter or gated-PMTs, but would require additional development. However, we now discuss these experiments in the revised manuscript as an important next step to understand these circuits. We discussed these experiments on page 15:

“Future experiments involving optogenetic perturbation of specific thalamic populations during simultaneous functional imaging of other thalamic pathways in awake mice performing tactile tasks will be essential to examine causal neural circuit mechanisms. One interesting approach might be to inject Cre and Flp expressing anterograde trans-synaptic viruses in Pr5 and Sp5 to be able to express different combinations of fluorophores, optogenetic actuators and activity reporters for more detailed analyses of specializations, correlations and interactions.”

In the revised manuscript, we also extended the discussion to frame the study in a broader context. It has been hypothesized that different types of thalamo-cortical inputs, core versus matrix, might play different roles based on their anatomical organization. Core-type axons are confined to small cortical domains and provide highly specific sensory information whereas matrix-type axons span a large horizontal portion of the cortex and seem to provide unspecific/untuned sensory information (Harris and Shepherd, 2015). A possible role of higher-order thalamic inputs could reside in their ability to alter the functional connectivity of large cortical networks, allowing different cortical areas involved in complementary computation to communicate under these conditions (Halassa and Kastner, 2017). In view of recent results regarding the role of POM cortical inputs in coupling top-down inputs in distal dendrites to somatic activity in L5 pyramidal neurons (Suzuki and Larkum, 2020), we hypothesize that the untuned whisker-evoked responses we observed could contribute to the facilitation of sensorimotor integration during specific time windows following stimulus onset. The contribution of POM-HO would, therefore, be contrasted to VPM-FO and POM-FO that provide specific sensory information to specialized cortical domains. We included this discussion in the revised manuscript on page 12:

“It has been hypothesized that different types of thalamocortical inputs, core versus matrix, might play different roles based on their anatomical organization. Core-type axons are confined to small cortical domains and provide highly specific sensory information whereas matrix-type axons span large horizontal domains of the cortex and seem to provide broadly-tuned sensory information⁴⁰. A possible role of higher-order thalamic inputs could reside in their ability to alter the functional connectivity of large cortical networks allowing different cortical areas involved in complementary computations to communicate under certain conditions⁴². In view of recent results suggesting a prominent role of layer 1 POM thalamocortical signals in coupling synaptic inputs in distal dendrites to somatic activity in L5 pyramidal neurons³³, we hypothesize that the untuned whisker-evoked responses we observed in POM-HO axons could contribute to facilitate sensorimotor integration in a context-dependent manner. The contribution of POM-HO could therefore be complementary to VPM-FO and POM-FO that provide specific sensory information to specialized distinct cortical domains.”

Minor comments:

1. Segregated axons from VPM vs intermingled axons of POm-FO: How did you define/quantify segregated vs intermingled?

We quantified axonal spatial distributions for VPM-FO and POm-FO tuned axons by separating all pixels within a field of view that show strong tuning to either B2 ($WSI < 0.75$) or C2 ($WSI > 0.75$) whisker deflections. We then used the correlation matrix of this pixel distribution to define a confidence interval with $p=90\%$ defining ellipses containing pixels responding to either whisker. We defined an overlap index as the surface of the intersection normalized by the average ellipse size between the two populations. This index is equal to 0 if the two ellipses are not intersecting and is equal to 1 if the two domains are identical. This index was significantly lower for VPM-FO axons distribution as opposed to POm-FO. We added this analysis in the method section and in the main text on page 7:

“We quantified the differences in the spatial distribution of highly-tuned axons for C2 and B2 whiskers using an index of overlap (see Methods). Comparing between VPM-FO and POm-FO thalamocortical projections, we found significantly more overlap between C2- and B2-responding axons in POm-FO than VPM-FO (VPM-FO: $n=19$ fields of view, 0.023 ± 0.014 mean \pm s.e.m.; POm-FO: $n=16$ fields of view, 0.217 ± 0.071 mean \pm s.e.m., Kruskal-Wallis unpaired test, $p=0.0063$).”

2. Fig 4d-f: how was the duration of the grey box (0.266sec) chosen? If I understand it correctly, it is longer than the duration of the whisker stimulation (0.2sec) but shorter than the average reaction time to lick (0.29sec).

The main criteria for this analysis was to choose a window that was shorter than the average reaction time to lick (0.29sec) but long enough to capture the calcium response elicited by whisker stimulation given that GCaMP6s has a slow rise time. We decided to use 0.266 sec to have a long enough window below the typical lick reaction time. Since our acquisition frame rate with the two-photon microscope (galvo-resonant mirror pair) was set around 30 frames/sec (corresponding to 0.033 sec/frame), 0.266 was the last frame before the average lick reaction time. The result of this analysis is robust with different windows, though. Below we illustrate the scatter plots comparing lick and no lick conditions for all thalamic populations using a window of 0.3 sec as requested by the reviewer in the next point 3 below. No differences were found with the results presented in the manuscript. In the revised manuscript, we explicitly point this out in the Results on page 8 and Analysis Methods sections on page 42.

3. On page 8, it is stated that the early phase of the VPM axon responses does not differ whether the mouse licks or doesn't lick the spout, but the calcium response latencies do indeed vary with first lick reaction time. These two seemingly contradicting statements are confusing to me. Is this why the cutoff for the "early phase" was set at 0.266sec? If it was set to the average reaction time to lick of 0.29sec, would you see a difference between lick vs no lick trials?

We thank the reviewer for pointing out the lack of clarity in the description of our analysis of lick response latency. Axonal calcium responses during lick trials typically contain two components. The first component is evoked by passive whisker stimulation and is locked to the onset of this stimulation. If calcium responses are evoked by lick-related events then a second component is present in the calcium response, for which the latency should vary with first lick reaction time (illustrated in panel a schematic below). In our analysis we try to capture the dominant calcium response component and identify whether it is locked on lick reaction time. To do so we compute the normalized cumulative distribution and measure the latency at 50%, which is well beyond the lick-independent response component (schematic in panel b). For axons in VPM-FO we can clearly identify these two components across trials (see Fig. 4b in the manuscript and panel c below). Because the typical lick reaction time is around

300 ms, there is no clear modulation of the response prior to this time point as indicated by the overlapping purple and black curves. However, a second component is present in the response which latency changes with reaction time. Using the analysis described above we could measure these latencies as illustrated for few examples in panels **d-e** corresponding trials highlighted with open squares in panel **c**. In contrast, in P_{Om} axons the dominant response corresponds to the passive whisker stimulation (see new Supplementary Figure 3 for instance) and its latency is independent of the reaction time. The strong modulation between lick and no lick trials precedes the first lick reaction time indicating different mechanisms. We improved the description of our analysis in the method section on page 43 to prevent further confusion.

Frangéul, L., Pouchelon, G., Telley, L., Lefort, S., Lüscher, C., and Jäbaudon, D. (2016). A cross-modal genetic framework for the development and plasticity of sensory pathways. *Nature* *538*, 96–98.

Halassa, M.M., and Kastner, S. (2017). Thalamic functions in distributed cognitive control. *Nat. Neurosci.* *20*, 1669–1679.

Harris, K.D., and Shepherd, G.M.G. (2015). The neocortical circuit: Themes and variations. *Nat. Neurosci.* *18*, 170–181.

Hubatz, S., Hucher, G., Shulz, D.E., and Férezou, I. (2020). Spatiotemporal properties of whisker-evoked tactile responses in the mouse secondary somatosensory cortex. *Sci. Rep.* *10*, 1–11.

Masri, R., Bezdudnaya, T., Trageser, J.C., and Keller, A. (2008). Encoding of stimulus frequency and sensor motion in the posterior medial thalamic nucleus. *J. Neurophysiol.* *100*, 681–689.

Minamisawa, G., Kwon, S.E., Chevée, M., Brown, S.P., and O'Connor, D.H. (2018). A Non-canonical Feedback Circuit for Rapid Interactions between Somatosensory Cortices. *Cell Rep.* *23*, 2718–2731.e6.

Ohno, S., Kuramoto, E., Furuta, T., Hioki, H., Tanaka, Y.R., Fujiyama, F., Sonomura, T., Uemura, M., Sugiyama, K., and Kaneko, T. (2012). A morphological analysis of thalamocortical axon fibers of rat posterior thalamic nuclei: A single neuron tracing study with viral vectors. *Cereb. Cortex* *22*, 2840–2857.

Pouchelon, G., Gambino, F., Bellone, C., Telley, L., Vitali, I., Lüscher, C., Holtmaat, A., and Jäbaudon, D. (2014). Modality-specific thalamocortical inputs instruct the identity of postsynaptic L4 neurons. *Nature* *511*, 471–474.

Suzuki, M., and Larkum, M.E. (2020). General Anesthesia Decouples Cortical Pyramidal Neurons. *Cell* *180*, 666–676.e13.

Urbain, N., Salin, P.A., Libourel, P.A., Comte, J.C., Gentet, L.J., and Petersen, C.C.H. (2015). Whisking-Related

Changes in Neuronal Firing and Membrane Potential Dynamics in the Somatosensory Thalamus of Awake Mice. *Cell Rep.* *13*, 647–656.

Wimmer, V.C., Bruno, R.M., De Kock, C.P.J., Kuner, T., and Sakmann, B. (2010). Dimensions of a projection column and architecture of VPM and POm axons in rat vibrissal cortex. *Cereb. Cortex* *20*, 2265–2276.

Reviewer #2 (Remarks to the Author):

The paper by El-Boustani et al. describes distinct thalamocortical pathways to the primary and secondary somatosensory cortices in the mouse. They distinguish three pathways of vibrissa information: a VPM-first order pathway (VPM-FO) that projects to S1, a POM-first order pathway (POM-FO) that projects principally to S2, and a POM-higher order (POM-HO) pathway that projects to layers 5a and 1 across S1 and S2. They also report how these pathways convey vibrissa information in awake behaving mice. The authors used an impressive variety of methods which include viral approaches, optogenetic stimulation combined with in vitro recordings, two-photon imaging with a microprism, and two-photon calcium imaging in the head-restrained mouse performing a go/nogo task upon vibrissa stimulation. Data are well illustrated and well quantified.

We thank the reviewer for his/her comments. We tried to address his/her concerns below.

Major point

Prior studies in rats (referred to in the paper) have identified three ascending pathways of vibrissa information; a lemniscal pathway (PrV-VPM-S1), a paralemniscal pathway (SpVlrostral-POM-S1,S2), and an extralemniscal pathway (SpVlcaudal-VPMvl-S2). It is not clear whether the POM-FO and POM-HO pathways correspond to the paralemniscal and extralemniscal pathways, respectively. The authors injected a very large amount of anteroAAV expressing Cre in the interopolaris nucleus (500 nl), which likely led to Cre expression in the thalamic targets of both the rostral and caudal divisions of the interopolaris nucleus (e.g., POM and VPMvl). This is a crucial point to clarify to avoid creating confusion in this field of research. As the anteroAAV expressing Cre has no reporter, the authors should consider co-injecting this virus with another AAV expressing a fluorescent reporter. Reducing the volume of injections (less than 100 nl) is mandatory to achieve specificity of labeling in the rostral and caudal aspects of the interopolaris nucleus. This is the main weakness of the paper, which questions the validity of the physiological results (though they are superbly illustrated).

Here, in our manuscript, we describe three distinct thalamic inputs to somatosensory cortices in mice: 1) Pr5 -> VPM-FO -> wS1:L4; 2) Sp5i -> POM-FO -> wS2:L4; and 3) Ctx:L5&6 -> POM-HO -> Ctx:L1&5A. Our results diverge from the classical view of 'lemniscal' and 'paralemniscal' pathways. 'Classically', we would assign the L1 and L5A axons from POM as 'paralemniscal', but our new data suggest that higher-order POM neurons do not receive paralemniscal (i.e. brainstem Sp5i) input. Instead our data suggest that the paralemniscal pathway (as defined by input from spinal trigeminal nuclei) provides direct sensory input predominantly to wS2, especially layer 4 neurons. Our data therefore fundamentally challenge the current typical descriptions of the whisker somatosensory signaling pathways.

Previous papers have highlighted that a part of POM neurons has direct projections in S2 L4 similar to the extralemniscal pathway (Frangeul et al., 2016; Minamisawa et al., 2018; Ohno et al., 2012; Pouchelon et al., 2014; Wimmer et al., 2010). In our experiments, we were using a double injection strategy to target specifically POM neurons through transsynaptic anterograde transfection. This required injection large amounts of viral vectors in Sp5i to ensure that enough neurons in the thalamus would express Cre-recombinase. Indeed this procedure was necessary to ensure that the Cre-off expression of eYFP would reveal neurons not expressing Cre-recombinase from the brainstem. The specificity of our expression in POM was obtained through a second smaller injection to express Cre-dependent fluorescent proteins in thalamic cells.

The reviewer points out previous studies highlighting interesting specializations of subdivisions of Sp5i, namely caudal and rostral parts. In our study, we did not differentiate between caudal and rostral subdivisions of Sp5i. However, in order to address the reviewer's concern, we performed a series of experiments where we targeted the rostral subdivision of Sp5i with a small amount of viral vectors (~50nl) to express the fluorescent protein tdTomato (see Figure below). We confirmed that axons from these neurons indeed innervate the anterior part of POM. Additionally we use the same injection coordinates for the dual injections (AAV1.CaMKII.Cre in SpVi-rostral and Cre-dependent tdTomato in POM). In these experiments, we still observed the same pattern of axonal innervation of POM-FO and wS2-L4. POM-FO and VPMvl might still merge posteriorly and thus be part of the same overall signaling pathway.

We now explicitly discuss all these points on page 11 and 12 of the manuscript:

"The Sp5i to POM-FO trigemino-thalamo-cortical circuit that we characterized resembles the previously described extralemniscal pathway of the rat in terms of axonal innervation in the cortex^{2,37} and whisker-related representation^{3,7}. The extralemniscal pathway arises from a caudal part of Sp5i projecting to the ventrolateral part of VPM (VPMvl), which in turn innervates layer 4 of wS2 and septal layer 4 domains of wS1. In our experiments, we did not distinguish between rostral and caudal subdivisions of Sp5i. However, in additional anatomical experiments, injections were targeted specifically to the rostral subdivision of Sp5i, which resulted in a

similar innervation pattern of POM-FO further projecting in wS2 L4. Further work is needed to investigate functional and anatomical neural circuit differences of caudal and rostral subdivision of Sp5i in mice. That the domains of POM and VPMv² receiving direct axonal inputs from Sp5i seem to merge in the caudal part of the thalamus³⁷ may help understanding of the organization of these pathways. In our anatomical characterization of the POM-FO pathway, we also sometimes found expression in neurons located in the outer periphery of the VPM, likely corresponding to VPMvl (Supplementary Fig. 1), potentially suggesting that POM-FO and VPMvl could be part of a common circuit. Our data thus suggest two major trigemino-thalamo-cortical pathways conveying parallel tactile information from the periphery to the thalamorecipient layer 4 of wS1 and wS2 with first-order synaptic properties³⁸, which challenges the classical hierarchal model of the whisker sensory system and suggests that wS2 can process sensory information independently from wS1³⁹.

The third nucleus that we could isolate is the complementary subdivision of POM, presumably receiving inputs only from the cortex as suggested in previous works¹⁷⁻¹⁹. Indeed, contrary to the classical view, these POM higher-order neurons do not appear to receive tactile inputs from the periphery (Fig. 1c), which could explain their functional responses displaying a lack of whisker-selectivity. Thalamocortical axons from these neurons are located predominantly in layer 1 and layer 5A and are of the matrix type spanning large regions of the cortex horizontally⁴⁰. Neurons in POM-HO are driven by the layer 5 and layer 6 of the cortex^{17,18,28} and could thus serve as a hub for cortico-thalamo-cortical communication, for example linking activity in wS1 and wS2⁴¹.

Figure. Anatomical projections from the rostral part of the interpolaris nucleus in the spinal trigeminal. **a**, Mice (n=5) were injected with viral vector to express tdTomato in all cells of the rostral part of Sp5i. After 2-3 weeks, brains were perfused and imaged with a two-photon serial tomography system. Each brain was then registered to the Allen mouse brain atlas. After normalization to the fluorescence in the injection site, all brains were averaged and resliced to obtain a sagittal view of the brain. The peak fluorescence in the brainstem was well localized in the rostral part of Sp5i. **b**, Axons originating from neurons in the rostral part of Sp5i targeted the anterior part of POM defined in the manuscript as POM-FO. **c**, To confirm that these axons project to the same cortical space than the extralemnisal pathway we injected a mouse with AAV1.CaMKII.Cre in the rostral part of Sp5i and Cre-dependent tdTomato in POM. We found a fluorescent population in POM-FO that projected in wS2 layer 4 confirming that the circuit that was classically described as paralemnisal with the following connectivity Sp5i-rostral->POM->S1 and S2 layer L1/L5a was instead directed to wS2 L4.

References

- Frangéul, L., Pouchelon, G., Telley, L., Lefort, S., Lüscher, C., and Jäbaudon, D. (2016). A cross-modal genetic framework for the development and plasticity of sensory pathways. *Nature* *538*, 96–98.
- Halassa, M.M., and Kastner, S. (2017). Thalamic functions in distributed cognitive control. *Nat. Neurosci.* *20*, 1669–1679.
- Harris, K.D., and Shepherd, G.M.G. (2015). The neocortical circuit: Themes and variations. *Nat. Neurosci.* *18*, 170–181.
- Hubatz, S., Hucher, G., Shulz, D.E., and Férézou, I. (2020). Spatiotemporal properties of whisker-evoked tactile responses in the mouse secondary somatosensory cortex. *Sci. Rep.* *10*, 1–11.
- Masri, R., Bezdudnaya, T., Trageser, J.C., and Keller, A. (2008). Encoding of stimulus frequency and sensor motion in the posterior medial thalamic nucleus. *J. Neurophysiol.* *100*, 681–689.
- Minamisawa, G., Kwon, S.E., Chevée, M., Brown, S.P., and O'Connor, D.H. (2018). A Non-canonical Feedback Circuit for Rapid Interactions between Somatosensory Cortices. *Cell Rep.* *23*, 2718–2731.e6.
- Ohno, S., Kuramoto, E., Furuta, T., Hioki, H., Tanaka, Y.R., Fujiyama, F., Sonomura, T., Uemura, M., Sugiyama, K., and Kaneko, T. (2012). A morphological analysis of thalamocortical axon fibers of rat posterior thalamic nuclei: A single neuron tracing study with viral vectors. *Cereb. Cortex* *22*, 2840–2857.
- Pouchelon, G., Gambino, F., Bellone, C., Telley, L., Vitali, I., Lüscher, C., Holtmaat, A., and Jäbaudon, D. (2014). Modality-specific thalamocortical inputs instruct the identity of postsynaptic L4 neurons. *Nature* *511*, 471–474.
- Suzuki, M., and Larkum, M.E. (2020). General Anesthesia Decouples Cortical Pyramidal Neurons. *Cell* *180*, 666–676.e13.
- Williams, M.N., Zahm, D.S., and Jacquin, M.F. (1994). Differential Foci and Synaptic Organization of the Principal and Spinal Trigeminal Projections to the Thalamus in the Rat. *Eur. J. Neurosci.* *6*, 429–453.
- Wimmer, V.C., Bruno, R.M., De Kock, C.P.J., Kuner, T., and Sakmann, B. (2010). Dimensions of a projection column and architecture of VPM and POm axons in rat vibrissal cortex. *Cereb. Cortex* *20*, 2265–2276.

Reviewers' Comments:

Reviewer #1:

Remarks to the Author:

The authors have now addressed all my previous comments and concerns. I appreciate the additional analyses they performed and their thoroughness. I have no further comments and support the publication of this manuscript.

I hope everyone is staying safe and well.

Reviewer #2:

Remarks to the Author:

The authors satisfactorily address my concerns. Overall this is an excellent paper, which will hopefully stimulate new interest on the role of the posterior group of the thalamus, a tote bag like thalamic region with ill-defined functions.